# Option Discovery via Differentiable Neural Decomposition

## Abstract

Option discovery via neural network decomposition is a promising way of discovering temporally extended actions in reinforcement learning. The challenge is that the number of sub-functions a network encodes grows exponentially with its size, so finding sub-functions that can be useful in downstream tasks is a difficult combinatorial search problem. In this paper, we turn this combinatorial search problem into a differentiable problem by showing that extracting sub-functions from a network is equivalent to learning masks over the neurons of the network. In addition to extracting sub-functions, we can also learn default input parameters to such sub-functions through masks over the inputs. Neuron masks select what to execute; input masks specify how to call it. We evaluate our masking scheme on grid-world problems with binary observations and on a robotics task with continuous action and observation spaces, using feedforward and recurrent policies. Our results show that masking can produce sub-functions with default input parameters that improve sample efficiency on downstream tasks.

## 1 Introduction

In transfer learning settings, an agent benefits from reusing skills learned in previous tasks. Knowledge reuse with neural networks can be complex due to factors such as catastrophic forgetting (French, 1999) and loss of plasticity (Dohare et al., 2024). Some of these difficulties do not arise when using other types of representations, such as programming languages. For example, DreamCoder learns a library of lambda calculus programs by compressing solutions to tasks it has solved, so that the programs in the library can reduce the complexity of solving downstream tasks (Ellis et al., 2023).

Alikhasi & Lelis (2024) took a programmatic view of neural networks to learn reusable neural sub-programs through a method called Dec-Options. Dec-Options decomposes neural networks encoding a solution to a sequential decision-making problem into an equivalent chain of if-then-else statements, known as a neural tree. Then, it evaluates each sub-tree of the neural tree for reusable sub-functions that could be stored in a library of programs to help reduce the complexity of downstream problems. These sub-functions were used as temporally extended actions, or options (Sutton et al., 1999).

The main drawback of Dec-Options is that it evaluates all possible sub-trees of a neural tree, whose size grows exponentially with the number of neurons in the original neural network. As a result, Dec-Options can only be used with small neural networks, which prevents it from being used on more challenging problems (Dec-Options were evaluated with networks with only six neurons). In this paper, we overcome this problem by searching in the space of possible sub-trees of the neural tree with gradient descent. Our method, Differentiable Dec-Options (DIDEC, which we pronounce "Dye-Deck"), leverages the fact that each sub-tree of the neural tree can be recovered by using a mask over the neurons of the underlying network. That way, masking neurons is equivalent to searching for sub-functions of the network, which allows us to search for reusable sub-functions in larger networks.

We also show that sub-functions needed to help the agent solve downstream tasks might only be recovered if we also learn "default input parameters" for the sub-functions. Consider the case where a robot learns a behavior that could be reused in different locations of the environment. However, the network encoding the behavior depends on features present only in the location where the agent learned it. In this case, a sub-function will encode the behavior that generalizes to any location only if we learn default location parameters, which allow the agent to "pretend" to be where it was when

it originally learned the behavior. DIDEC also learns default parameters with gradient descent by masking the input observation—it searches for sub-functions and their default parameters.

We evaluate DIDEC in a transfer learning setting where we have neural models encoding solutions to previous tasks, and the agent can use these models to improve its sample efficiency while learning how to solve the current task. DIDEC decomposes the networks encoding the solution to previous problems, equipping the agent with a library of sub-functions, or options. We hypothesize that DIDEC can extract reusable sub-functions from networks larger than those used in previous work. We evaluate this hypothesis by checking whether DIDEC's options can generalize better to downstream tasks than baselines that use the solution to previous tasks without decomposing them. Empirical results in problems with complete and partial observability support our hypotheses.

## 2 PROBLEM FORMULATION

We are interested in solving partially observable Markov decision processes (POMDP), which are defined as $(S, A, O, p, q, r, S_0)$. Here, $S$, $A$, $O$ are the sets of states, actions, and observations, respectively. The function $p : S \times A \to S$ defines the transition dynamics of the environment by returning the next state $s_{j+1}$ given the current state $s_j$ and an action the agent takes at $s_j$; function $q : S \to O$ defines what the agent observes in the current state. Function $r : S \times A$ defines the reward value the agent observes after performing an action in a given state. Finally, $S_0$ defines the distribution of initial states. A policy $\pi : O \times A \to [0, 1]$ receives an observation $o$ and an action $a$ and returns the probability that $a$ is taken in $o$. We consider the transfer learning setting where we learn to solve a sequence of POMDPs $P_1, P_2, \cdots, P_{i-1}$ and is evaluated in the $i$-th POMDP, $P_i$. This means that for each $P_j$ with $j < i$, we have a policy $\pi_j$ that approximates $\pi_j^*$, where

$$\pi_j^* = \arg\max_{\pi \in \Pi} \mathbb{E}_{s_0 \sim S_0}[R(s_0, \pi)].$$

In an episodic setting with $T$ time steps, $R(s_0, \pi) = \sum_{t=0}^{T} r(s_t, a_t)$, and $\Pi$ is the class of possible policies. Given the collection of neural policies $\{\pi_j\}_{j=1}^{i-1}$, we want to approximate $\pi_i^*$ while optimizing for sample efficiency. We approach this problem by decomposing the neural policies $\{\pi_j\}_{j=1}^{i-1}$ (Section 3) into options (Section 4) that can improve the agent's sample efficiency while solving $P_i$.

## 3 DECOMPOSING FEEDFORWARD NEURAL NETWORKS

We assume that the policies $\{\pi_j\}_{j=1}^{i-1}$ are encoded in fully connected feedforward neural networks, such as the one shown in the top-right corner of Figure 1. Each layer $k$ has $n_k$ neurons $(1, \cdots, n_k)$, with $n_1 = |X|$, where $X$ is the observation vector passed as input to the network. The network's trainable parameters are the values between any subsequent layers $k$ and $k + 1$. We denote such weights as $W^k \in \mathbb{R}^{n_{k+1} \times n_k}$ and $B^k \in \mathbb{R}^{n_{k+1}}$. The $r$-th row of $W^k$, denoted $W_r^k$, together with the $r$-th entry of $B^k$, denoted $B_r^k$, represent the weights and bias term of the $r$-th neuron of the $(k + 1)$-th layer. We denote the vector with the values produced in the $k$-th layer of the network as $A^k \in \mathbb{R}^{n_k}$. Here, $A^1 = X$ and $A^m$ is the model output of a network with $m$ layers. We compute $A^k = g(Z^k)$, where $g(\cdot)$ is an activation function, and $Z^k = W^{k-1}A^{k-1} + B^{k-1}$. We consider piecewise-linear activation functions, such as ReLU, where $g(z) = \max(0, z)$, to enable network decomposition.

Alikhasi & Lelis (2024) showed how mapping a neural network that uses piecewise-linear activation functions to an equivalent neural tree enables the extraction of sub-functions from the network. We explain this mapping with the example in Figure 1 (c) and (d). Each neuron is mapped to a level in the tree. In Figure 1, neuron $A_1^2$ is represented by the root level of the tree, $A_2^2$ by the second level of the tree, and $A_1^3$ by the tree's leaves. Each node considers the two possible outcomes of a ReLU neuron: if $Z \leq 0$, the output is 0.0 (left branch); if $Z > 0$, the output is $Z$ (right branch). In our example, the output of the leaf nodes in the left sub-tree of the root is calculated for $A_1^2 = 0$, while the output of the leaf nodes in the right sub-tree of the root is calculated for $A_1^2 = Z_1^2$. For example, when following the left branch twice from the root, we have $A_1^2 = A_2^2 = 0$, so $A_1^3 = \sigma(5)$, which is the Sigmoid function of the bias term of the output neuron. If we follow the right and then the left branches from the root, then $A_1^2 = -x_1 + 4x_3$ and $A_2^2 = 0$, so the leaf output is $\sigma(-1(-x_1 + 4x_3) + 5) = \sigma(x_1 - 4x_3 + 5)$. Each sub-tree of the neural tree provides a sub-function of the policy that can be potentially reused to improve sample efficiency in downstream problems.

# 4 DEC-OPTIONS

Dec-Options extracts sub-functions of the policies in $\{\pi_j\}_{j=1}^{i-1}$ so that they can be used as options (Sutton et al., 1999). An option $\omega$ is a tuple $(I_\omega, \pi_\omega, T_\omega)$, where $I_\omega$ is the set of observations in which the option can be invoked, $\pi_\omega$ is the policy that the agent follows once the option starts, $T_\omega$ is a function that receives an observation $o_t$ and returns the probability that the option terminates in $o_t$. We consider the call-and-return execution of options, where the agent follows $\pi_\omega$ until $\omega$ terminates.

Dec-Options operates in two steps, given the policies $\{\pi_j\}_{j=1}^{i-1}$, which we will refer to as $\Pi_{\text{train}}$. First, it decomposes each policy $\pi_j$ in $\Pi_{\text{train}}$ into all possible sub-trees of the neural tree of $\pi_j$ (Section 4.1). Then, it selects a subset of these sub-trees to be used as options for solving $P_i$ (Section 4.2).

## 4.1 SUB-TREES TO OPTIONS

Options are temporal abstractions because they encode policies that execute over multiple steps before terminating. Feedforward neural networks do not represent programs with loops, which can be run many iterations; instead, they represent chains of if-then-else structures. Dec-Options incorporates the temporal aspect of these abstractions by wrapping each sub-tree, extracted from a policy, in loops.

Let $\mathcal{T}_{\text{train}} = \{\mathcal{T}_j\}_{j=1}^{i-1}$ be the set of trajectories obtained by rolling each $\pi_j$ in $\Pi_{\text{train}}$ out from an initial state of each POMDP in $\{P_j\}_{j=1}^{i-1}$. Each trajectory $\mathcal{T}_j$ is a sequence of observation-action pairs of the form $\{(o_0, a_0), (o_1, a_1), \cdots, (o_{T_j}, a_{T_j})\}$. Let $T_{\max} = \max_j T_j$ be the length of the longest trajectory in $\Pi_{\text{train}}$, and $U_j$ be all sub-trees that can be extracted from $\pi_j$. Dec-Options wraps each $u$ in $U_j$ in programs `repeat(t):u`, where sub-tree $u$ is invoked $t$ times before termination. Dec-Options considers one such program for each $t$ in $\{2, \cdots, T_{\max}\}$. The programs obtained from the sub-trees in $U_j$ form the set of options $\Omega_j$ extracted from $\pi_j$. For Dec-Options, $I_\omega = O$, that is, the options can be invoked from any observation, and $T_\omega$ is deterministic as options are executed for $t$ steps.

## 4.2 DEC-OPTIONS SUBSET SELECTION

We denote the set of options extracted from $\Pi_{\text{train}}$ as $\Omega = \cup_{j=1}^{i-1} \Omega_j$. Given the size of $\Omega$, attempting to use all options to solve downstream problems would slow down learning, as the agent would have to learn to use a very large number of options. That is why Dec-Options selects a subset $\Omega'$ of $\Omega$ based on the Levin loss (Orseau et al., 2018) of $\Omega'$. The Levin loss approximates the usefulness of $\Omega'$ in solving downstream tasks. Intuitively, the Levin loss evaluates whether the likelihood of an agent solving a problem $P_j$ would increase if we augmented the action space of the agent with $\Omega'$.

$$\mathcal{L}(\mathcal{T}_j, \pi) = \frac{|\mathcal{T}_j|}{\prod_{(o,a) \in \mathcal{T}_j} \pi(o, a)} \,. \tag{1}$$

The Levin loss $\mathcal{L}(\mathcal{T}_j, \pi)$ computes the expected number of samples an agent following $\pi$ requires to recover the trajectory $\mathcal{T}_j$. The denominator of $\mathcal{L}$ gives the expected number of rollouts the agent following $\pi$ needs to perform to observe the trajectory $\mathcal{T}_j$; the numerator is the required number of agent interactions with the environment in each rollout. Dec-Options selects a subset $\Omega'$ of $\Omega$ such that the resulting policy $\pi$ minimizes the Levin loss on trajectories $\mathcal{T}_{\text{train}}$.

To simulate the scenario in which the agent is starting to learn how to solve a problem, the policy $\pi$ used in the computation of the Levin loss is the uniform policy—a randomly initialized neural network can produce a probability distribution over actions that is close to uniform. The uniform policy on the agent's action space augmented with options $\Omega'$ is denoted $\pi_u^{\Omega'}$. In this way, the larger the set $\Omega'$, the smaller $\pi(o, a)$, which increases the Levin loss. Conversely, if the options in $\Omega'$ cover sub-trajectories of the trajectories in $\mathcal{T}_{\text{train}}$, then the number of decisions the agent must make to reproduce the trajectories is reduced, which decreases the Levin loss. The loss balances the negative (decrease of $\pi(o, a)$) and positive (decrease in the number of decisions) effects of growing $\Omega'$.

Dec-Options approximates a solution to the problem of selecting a subset $\Omega'$ of $\Omega$ that minimizes the sum of the Levin loss of the trajectories in $\mathcal{T}_{\text{train}}$. Importantly, when computing the value of $\mathcal{L}(\mathcal{T}_j, \pi_u^{\Omega'})$, Dec-Options does not consider the options $\omega$ in $\Omega'$ that were extracted from $\pi_j$. This is to prevent the selection of trivial policies that do not generalize to downstream problems. For example, the policy that reduces the Levin loss the most for the trajectory $\mathcal{T}_j$ is $\pi_j$, which is unlikely

to generalize because it is too specialized in $\pi_j$. Alikhasi & Lelis (2024) used a greedy algorithm to approximate a solution to the subset selection problem. They also presented a dynamic programming procedure to compute the Levin loss for a given subset $\Omega'$. Please refer to Appendix A for details.

## 5 DIFFERENTIABLE DEC-OPTIONS (DIDEC)

Dec-Options has two important shortcomings. First, since the number of sub-trees grows exponentially with the number of neurons in the network, Dec-Options can only be used with small networks (Alikhasi & Lelis (2024) used networks with only six neurons), limiting the approach's applicability. Second, as we show in Section 5.3.1, the functions Dec-Options extracts from neural networks might not generalize to downstream problems because the extracted functions do not come with "default parameters". Differentiable Dec-Options (DIDEC) overcomes these two shortcomings.

### 5.1 SUB-TREE EXTRACTION AS MASKING NEURONS

Alikhasi & Lelis (2024) showed that the number of possible sub-trees a neural network with $d$ hidden neurons is $\sum_{i=0}^{d} \binom{d}{i} \cdot 2^i$. For the neural tree in Figure 1, we have $1 + 4 + 4 = 9$ sub-trees. The $1$ represents the entire neural tree, the first $4$ represents the sub-trees of the root: there are $2$ sub-trees when $A_1^2$ is the root, and another $2$ when $A_2^2$ is the root. Finally, the last $4$ is the number of leaf nodes.

We note the binomial identity $(1+x)^d = \sum_{i=0}^{d} \binom{d}{i} \cdot x^i$, so if $x = 2$, then $\sum_{i=0}^{d} \binom{d}{i} \cdot 2^i = (1+2)^d = 3^d$. This identity is useful because it suggests a gradient-based solution to the problem of finding helpful sub-functions. In a sub-tree of the neural tree, each ReLU neuron can be **active**, when it returns $z$, **inactive**, when it returns $0$, or **part of the sub-function**, when the neuron's function is accounted for in the sub-tree. To illustrate, consider the left sub-tree of the tree in Figure 1, neuron $A_1^2$ is inactive (we follow its left child), while $A_2^2$ is part of the program because its node is in the sub-tree. This means that extracting a sub-tree from the neural tree is equivalent to setting one of the following states to each neuron: active, inactive, or part of the sub-function, for $3^d$ possibilities.

DIDEC masks neurons to extract sub-trees of the underlying tree. Masks are given by a matrix

$$\Theta^k \in \mathbb{R}^{n_k \times 3} = \left( (\theta_{1,1}^k, \theta_{1,2}^k, \theta_{1,3}^k), (\theta_{2,1}^k, \theta_{2,2}^k, \theta_{2,3}^k), \ldots, (\theta_{n_k,1}^k, \theta_{n_k,2}^k, \theta_{n_k,3}^k) \right)$$

for the $n_k$ neurons in the $k$-th layer of the network. The parameters $(\theta_{j,1}^k, \theta_{j,2}^k, \theta_{j,3}^k)$ are used in a Softmax operation to determine the state of the $j$-th neuron in the $k$-th layer, as shown in Algorithm 1.

---

**Algorithm 1** MASKED FORWARD PASS OF THE $k$-TH LAYER

1: $Z^k \leftarrow W^{k-1} A^{k-1} + B^{k-1}$
2: **for** $i = 1$ **to** $n_k$ **do**
3:     $p_i \leftarrow \text{Softmax}(\Theta_{i,:}^k) \in \mathbb{R}^3$
4:     $M_{i,:}^k \leftarrow \text{OneHot}(p_i) \in \{0,1\}^3$
5: $A^k \leftarrow M_{:,1}^k \odot \mathbf{0} + M_{:,2}^k \odot Z^k + M_{:,3}^k \odot \text{ReLU}(Z^k)$

---

In line 1 of Algorithm 1, we compute the logits $Z^k$. In lines 2 to 4, we obtain the mask $M^k$ by applying Softmax to each row of $\Theta^k$ and discretizing to a one-hot vector. The $j$-th element of row $i$ is $1$ if it is the largest entry, and $0$ otherwise. Finally, in line 5 we compute the masked output $A^k$, where $\mathbf{0} \in \mathbb{R}^{n_k \times 1}$ denotes the zero vector: rows with a $1$ in the first column yield $0$ (inactive), in the second column yield $Z^k$ (active), and in the third column yield $\text{ReLU}(Z^k)$ (part of the sub-program).

The function OneHot in line 4 is non-differentiable due to its implicit max operation. When updating $\Theta^k$ with gradient descent, we use the straight-through estimator (Bengio et al., 2013), which passes the gradient computed up to the max operation in the backward pass directly to the Softmax layer. We also considered the modified tanh function of Pitis (2017) and Koul et al. (2019) to discretize over 3 values, but preliminary experiments favored the use of the Softmax approach of Algorithm 1.

### 5.1.1 LEARNING MASKS AS NEURAL SUB-TREE SELECTION

Ideally, we would learn the parameters $\Theta^k$ such that we minimize the Levin loss of a subset of options $\Omega'$. However, the Levin loss is computed with Dec-Options's dynamic programming process, as

shown in Appendix A, which is not differentiable. Instead of optimizing for the Levin loss directly, DIDEC uses a behavioral cloning approach (Schaal, 1997) to discover options and optimizes for the cross-entropy loss of sub-trajectories of $\mathcal{T}_{\text{train}}$ to learn a set of masks encoding sub-programs. We then select a subset of such a set while minimizing the Levin loss (see Appendix A).

For each $\mathcal{T}_j$ of $\mathcal{T}_{\text{train}}$, we consider all sub-trajectories $\tau$ of observation-action pairs of $\mathcal{T}_j$ with length $z$ in $\{2, 3, \cdots, z_{\max}\}$. Here, $z_{\max} \leq T_j$ is a hyperparameter. Each sub-trajectory is used to train masks for neurons of the $\pi_i$ network, where $i \neq j$. We ensure that $i \neq j$ to allow for generalization, as it attempts to extract sub-functions of $\pi_i$ that help solve a problem $P_j$ for which $\pi_i$ was not trained to solve. We use the cross-entropy loss to learn $\Theta$ such that the masked network predicts the actions in the observation-action pairs of $\tau$. We hypothesize DIDEC's gradient-based process for selecting sub-trees of the policy allows for the discovery of options that generalize to downstream problems.

For a sub-trajectory $\tau$ with length $b$ of $\mathcal{T}_j$, we train the parameters $\Theta$ to generate an option of the form `repeat(b):`$\pi_j^\Theta$, where $\pi_j^\Theta$ is the policy $\pi_j$ masked with $\Theta$, as shown in Algorithm 1. Larger $z_{\max}$ values will result in the generation of more options. However, options trained with longer sub-trajectories tend to be specific to the behavior needed to solve the problem from which the sub-trajectory was extracted, thus they are less likely to generalize to downstream problems. DIDEC's set $\Omega$ is formed by one option for each sub-trajectory $\tau$. The process of selecting a subset of options that minimizes the Levin loss is NP-hard in general (Appendix D). Alikhasi & Lelis (2024) used a greedy approach to approximate a solution to the problem. Preliminary experiments showed that this greedy approximation often returned solutions that were far from the optimal solution. Instead, we use a stochastic hill climbing approach to approximate a solution to the subset selection problem for both DIDEC and baselines. We present the stochastic hill climbing search in Appendix B.

## 5.2 DIDEC'S RUNNING TIME

DIDEC considers $\sum_{i=2}^{z_{\max}} (T_{\max} - i + 1)$ sub-trajectories per trajectory in $\mathcal{T}_{\text{train}}$, totaling $O(T_{\max} z_{\max} | \mathcal{T}_{\text{train}}|)$ sub-trajectories. Each sub-trajectory is used to train one option for each base policy in $\Pi_{\text{train}}$ (recall that $|\Pi_{\text{train}}| = |\mathcal{T}_{\text{train}}|$). Since training requires $E$ gradient steps in a network with $P$ trainable weights, the complexity of generating DIDEC's option set $\Omega$ is $O\left(T_{\max} z_{\max} |\mathcal{T}_{\text{train}}|^2 E P\right)$. After generating all options, DIDEC performs a stochastic hill-climbing procedure to select a subset. Although optimal subset selection is NP-hard in general (Appendix D), the approximation in Algorithm 3 (lines 20–34) is efficient in practice. Treating $z_{\max}$, $E$, and $P$ as fixed hyperparameters that do not scale with the input size, the option-generation stage simplifies to $O\left(T_{\max} |\mathcal{T}_{\text{train}}|^2\right)$. Thus, the computational cost of learning options is quadratic in the number of base trajectories (and the policies that generated them) and linear in the length of those trajectories. The number of base trajectories is small relative to their length, and policy training itself requires many more interactions with the environment than DIDEC's option-learning stage. Importantly, DIDEC's time complexity does not include the $3^d$ term of Dec-Options, for searching in the space of sub-programs of a network.

## 5.3 LEARNING DEFAULT PARAMETERS

While sub-trees of a neural tree might encode helpful sub-functions that can be reused in downstream tasks, in this section we argue that neural decomposition can be more effective if we learn "default parameters" to the extracted sub-functions. Consider the motivating example in the next section.

### 5.3.1 MOTIVATING EXAMPLE

Consider the ComboGrid problem shown in the upper left corner of Figure 1. The cell in the $2 \times 2$ grid are denoted as $x_1$, $x_2$, $x_3$, and $x_4$; the agent 'A' starts in $x_3$. In ComboGrid, the agent needs to perform a sequence of actions until the effect of moving to a different cell is observed. The agent can perform two actions in a given time step: $0$ or $1$. After performing action $1$ and then $0$, the agent moves to $x_1$; the "combo" $1, 0$ produces the effect of moving up. Similarly, the sequence $1, 1$ produces the effect of moving right. The sequence $1, 0, 1, 1$ moves the agent from $x_3$ to $x_2$.

Observations are given by a one-hot encoding of the position of the agent on the grid and an extra bit, $x_5$, which indicates whether the number of actions the agent has taken so far is even ($x_5 = 1$) or odd ($x_5 = 0$). This way, if $x_5 = 1$, then the agent has not started a combo sequence. The observation of the agent shown in the grid is given by $o_1$ in the table of observation-action pairs. Observations $o_2$

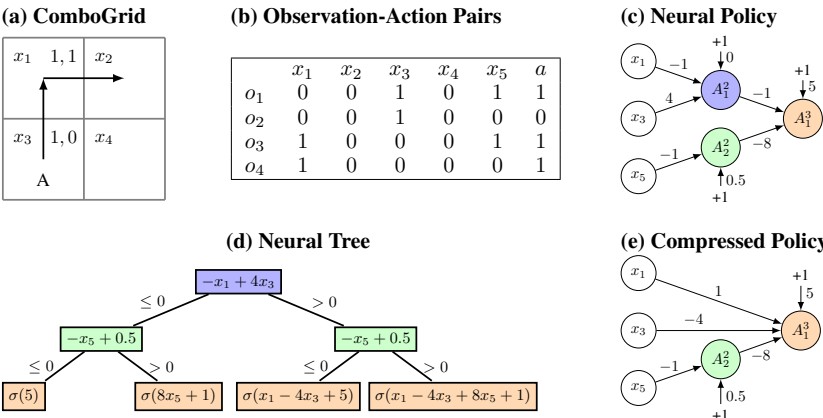

Figure 1: **(a)** Instance of a ComboGrid environment with combos of length two. The sequence $1, 0$ moves up, while $1, 1$ moves right. Agent 'A' starts at $x_3$ and reaches $x_2$ after $1, 0, 1, 1$. **(b)** Observation-action pairs for the trajectory in (a). The observation includes agent location and bit $x_5$, set to $1$ if no action was taken in a sequence, and $0$ otherwise. **(c)** Neural network that fits the data in (b), with ReLU hidden units and a sigmoid output; zero-weight connections are omitted. **(d)** Neural tree equivalent to (c): the left sub-tree encodes the "right combo" and the right sub-tree with default $x_3 = 1$ encodes the "up combo". **(e)** Compressed model obtained by setting neuron $A_1^2$ active.

and $o_3$ are obtained by applying actions $1, 0$ from $o_1$. Finally, once the agent performs action $1$ in $o_4$, it moves to cell $x_2$, whose observation is not shown in the table of observation-action pairs.

The neural network shown in Figure 1 (c) produces the sequence of actions given in the table of observation-action pairs. This network uses ReLU actions in the hidden layer and a sigmoid function in the output layer. The neural tree shown in Figure 1 (d) is equivalent to the neural network. The left sub-tree of the neural tree encodes the "right combo". The nodes in it depend only on $x_5$, so the sub-function it represents is independent of the agent's location. The root of the sub-tree checks whether $-x_5 + 0.5 \leq 0$. Since before starting the sequence, $x_5 = 1$, we follow the left child, leading to $\sigma(5) = 0.99$ (action 1). After action 1 is performed, $x_5 = 0$, so we follow the right child to the function $\sigma(8x_5 + 1) = \sigma(1) = 0.73$ (action 1). Equipping the agent with the left sub-tree of the neural tree can help solve downstream problems, for it encodes one of the combos of the domain.

Consider now the right sub-tree of the neural tree. Before the agent starts performing a sequence, $x_5 = 1$, so we follow the left child, leading to $\sigma(x_1 - 4x_3 + 5)$. Note that $x_1 - 4x_3 + 5 \geq 1$ for any combination of $x_1$ and $x_3$, so $\sigma(x_1 - 4x_3 + 5) \geq 0.73$, thus always producing action 1 when $x_5 = 1$. After performing action 1, $x_5 = 0$ so we follow the right child, leading to $\sigma(x_1 - 4x_3 + 8x_5 + 1) = \sigma(x_1 - 4x_3 + 1)$. In this case, the output value the model produces depends on both $x_1$ and $x_3$, and if $x_3 = 1$, then $x_1 - 4x_3 + 1 \leq -2$ and $\sigma(x_1 - 4x_3 + 1) \leq 0.11$. Thus, this sub-tree always produces action 0 when $x_5 = 0$ and $x_3 = 1$. We can only obtain the "up combo" independent of the agent location if we fix $x_3 = 1$, so it can help solve downstream problems. DIDEC can extract the right sub-tree by masking the neurons and, simultaneously, learn the default parameter $x_3 = 1$, so the agent can "pretend" to be in $x_3$ to perform the "up combo" from any cell of the grid.

### 5.3.2 LEARNING INPUT MASKS

In addition to masking neurons, DIDEC also masks input values to allow for default input parameters. We consider learning masks for input values for problem domains with discrete and finite inputs. For binary inputs, we learn masks with the same Softmax approach described in Section 5.1. The input masks define one of the three possibilities for a given binary input: **always 1**, **always 0**, or **read value from environment**. For example, a generalizable "up combo" can be obtained by setting the mask of $x_3$ to "always 1", while all other input values could be "read value from the environment".

For continuous features, we learn a decision between using a learned input feature or reading the input feature from the environment. Similarly to Algorithm 1, for the $i$-th input feature we compute

$$M_{i,:} \leftarrow \text{OneHot}(\text{Softmax}(\Theta_{i,:})) \in \{0, 1\}^2$$

The inputs are then computed with $M$ to use either a learned value $g(\Theta_m)$ or the original input $X$

$$M_{:,1} \odot g(\Theta_m) + M_{:,2} \odot X.$$

The parameters $\Theta_{i,:}$ in $\mathbb{R}^2$ for each input $i$ represent the decision of whether to use a learned input value or to read it from the environment. The parameters $\Theta_m$ in $\mathbb{R}^{n_1}$ represent the learned values, one for each of the $n_1$ input values. The mask $M$ in $\mathbb{R}^{n_1 \times 2}$ decides which one to use: $g(\Theta_m)$ or $X$. The function $g$ is chosen according to range of input values. For example, in our experiments on Meta-World (Yu et al., 2021), the input values are in $[-1, 1]$, so in this domain we use $\tanh$ as $g$.

Note that masking input values might be sufficient to learn sub-functions that generalize. For example, learning that $x_3$ is always 1 in the problem of Figure 1 is equivalent to extracting the right sub-tree with the default parameter $x_3 = 1$, thus potentially making the masking of neurons unnecessary. We evaluate DIDEC with three masking schemas: input-only, neurons-only, and input-and-neurons.

Independent of the benefit of masking neurons in discovering options, masking neurons can be a valuable compression scheme. In the example from Figure 1 (e), once we mask $A_1^2$ to be active, we can reduce the size of the network by making $A_1^3$ a function of $x$ and $A_2^2$: $A_1^3 = \sigma(x_1 - 4x_3 - 8A_2^2 + 5)$ and removing $A_1^2$ from the model. In general, every neuron at layer $k$ that is masked to be active or inactive can be removed by rewriting the function of the neurons at layers $k + 1$ accordingly. This compression is similar to how a software engineer extracts functions from a codebase: both the extracted functions and the masked models have fewer lines than the original implementation.

The approach of masking only the inputs has the advantage of being applicable to neural networks that use activation functions other than piecewise-linear functions and to other neural architectures. We evaluate DIDEC with input-only masking to extract options from $\tanh$ recurrent networks.

### 5.3.3 MASKING RECURRENT MODELS

In policies represented by recurrent models such as LSTMs (Hochreiter & Schmidhuber, 1997) and GRUs (Chung et al., 2014; Bahdanau et al., 2015), in addition to masking neurons and input values, we must also mask the initial hidden states. Recurrent models can be viewed as approximating finite state machines (FSMs), where the hidden state represents the current mode of the underlying FSM. When extracting sub-functions from a recurrent model, DIDEC must therefore specify the initial mode of the FSM. This is achieved by learning the initial hidden state together with DIDEC's masks. For the initial hidden vector $\mathbf{h_0} \in [-1, 1]$, DIDEC learns parameters $\Theta_h$ and sets $\mathbf{h_0} = \tanh(\Theta_h)$, with $\Theta_h$ of the same shape as $\mathbf{h_0}$. For LSTMs, we also learn the initial memory vector $\mathbf{c_0}$, which is unbounded. For numerical stability, DIDEC parameterizes $\mathbf{c_0}$ with two tensors $\Theta_s$ and $\Theta_m$ of the same shape as $\mathbf{c_0}$, representing sign and magnitude, respectively, and sets

$$\mathbf{c_0} = \tanh(\Theta_s) \odot \text{SoftPlus}(\Theta_m).$$

Appendix C presents DIDEC's pseudocode, which summarizes all its steps for discovering options.

## 6 RELATED WORK

**Options** Early work relied on manually designed options (Sutton et al., 1999), but many methods now learn options automatically—though they often require human choices such as the number of options (Bacon et al., 2017; Igl et al., 2020) or their duration (Frans et al., 2017; Tessler et al., 2017). DIDEC avoids such supervision but depends on data from previously solved tasks. While options have also been explored for improving exploration (Jinnai et al., 2020; Machado et al., 2023) and faster credit assignment (Mann & Mannor, 2014), DIDEC focuses on extracting functions that encode reusable behaviors to facilitate downstream knowledge transfer (Konidaris & Barto, 2007).

**Transfer Learning** Knowledge transfer across tasks has been studied via regularization (Kirkpatrick et al., 2017), architectural priors (Rusu et al., 2016; Yoon et al., 2017; Schwarz et al., 2018), and experience replay (Rolnick et al., 2019). A common strategy is to reuse parts of pretrained models (Clegg et al., 2017; Shao et al., 2018). Unlike these approaches, DIDEC transfers knowledge by extracting reusable sub-programs (options) through gradient-based network decomposition.

**Library Learning**   DIDEC is a library-learning method (Cao et al., 2023; Bowers et al., 2023; Rahman et al., 2024; Palmarini et al., 2024). As a representative example, DreamCoder (Ellis et al., 2023) builds a library of reusable lambda calculus functions to solve supervised learning problems. Similarly, DIDEC constructs a library of reusable programs, but the programs are extracted from neural networks and the underlying problem is reinforcement learning, and not supervised learning. DIDEC contributes toward bridging symbolic and neural paradigms in library-learning methods.

**Masking Networks**   Masking has been used in transfer learning, e.g., SupSup (Wortsman et al., 2020) and Modulating Masks (Ben-Iwhiwhu et al., 2022), but these approaches mask weights, not neurons or input features. In contrast, DIDEC masks neurons with piecewise-linear activations, allowing for the extraction of sub-functions. Input masking has also been explored—for mitigating visual distractions (Bertoin et al., 2022; Grooten et al., 2024) or learning auxiliary tasks (Yu et al., 2022)—but DIDEC uses input masking to define default parameters to enable generalization.

## 7   EMPIRICAL EVALUATION

We evaluate our hypothesis that DIDEC can extract options that generalize to downstream tasks, even when using networks much larger than the six-neuron networks used by Dec-Options.

We use Advantage Actor-Critic (A2C) (Mnih et al., 2016) and Proximal Policy Optimization (PPO) (Schulman et al., 2017) with either a feedforward network encoding the policy or an LSTM, depending on whether the problem is partially observable or not. Our network sizes range from 64 to 256 neurons, resulting in a range of $3.43 \times 10^{30}$ to $1.39 \times 10^{122}$ sub-trees, which far exceeds the capacity of enumerative approaches. The architectures we use are detailed in Appendix G.

We consider sets of problems $\{P_j\}_{j=1}^i$ that have observations of the same shape, but differ in terms of features and reward function. We performed experiments on MiniGrid (Chevalier-Boisvert et al., 2023), MiniHack (Samvelyan et al., 2021), ComboGrid (Alikhasi & Lelis, 2024), and Meta-World (Yu et al., 2021). We use A2C in our MiniGrid experiments and PPO in MiniHack, ComboGrid, and Meta-World. Our choice of learning algorithm is arbitrary and it illustrates the applicability of DIDEC. Neuron masking can be used if the activation functions are piecewise linear, and input masking can be used if the input domains are finite, which includes continuous inputs as in Meta-World. We use feedforward networks with MiniGrid, MiniHack, and Meta-World, and an LSTM with ComboGrid.

**MiniGrid.**   We consider eight Simple-Crossing configurations as training problems and FourRooms as the test problem. We also use three MultiRoom environments for training and a fourth for testing. MultiRoom is more challenging than FourRooms because the agent must find a key and unlock a door, and because room shapes vary, making generalization harder. We adopt MiniGrid's original action space (turn left, right, move) and an egocentric $9 \times 9$ observation window.

**MiniHack.**   We consider the corridors challenge of MiniHack, where the agent learns to navigate in long procedurally generated rooms and corridors of the game NetHack (Küttler et al., 2020). This environment offers dynamics and challenges different from MiniGrid, such as the agent starving. We consider four environments as training, and a fifth environment with longer corridors and more rooms as the test problem. As in MiniGrid, we also use an egocentric view of size $9 \times 9$ around the agent.

**ComboGrid.**   In ComboGrid, the agent must learn action sequences whose effects occur only after completing an "action combo". We consider three primitive actions (0, 1, 2) and three combos: 0,0,1 (turn right); 1,1,0 (turn left); and 2,2,1 (move forward). Because observations change only after a combo, the agent must remember sequences. Training uses four $5 \times 5$ grids surrounded by walls, leaving a $3 \times 3$ area where the agent collects a marker. For testing, we remove the walls to yield a full $5 \times 5$ grid with four markers. We study two reward settings: dense, where the agent receives a positive reward per marker, and sparse, which rewards only after collecting all markers.

**Meta-World.**   In Meta-World, the agent controls a robotic arm to perform various tasks. As observations, the agent receives the coordinates of different objects in 3D space; as actions, it can control the direction of the robotic arm's gripper, as well as opening and closing it. Both observations and actions are continuous. In our experiments, we consider two training problems where the agent moves a plate into a hockey net. The training problems differ in the position of the arm, plate, and net. The test problems are more difficult as the distance between the plate and the net changes, one by 0.85 times the distance in the training problems (Closer Goal) and the other by 1.35 times the distance (Farther Goal). Since selecting a subset of options while minimizing the Levin loss is inherently

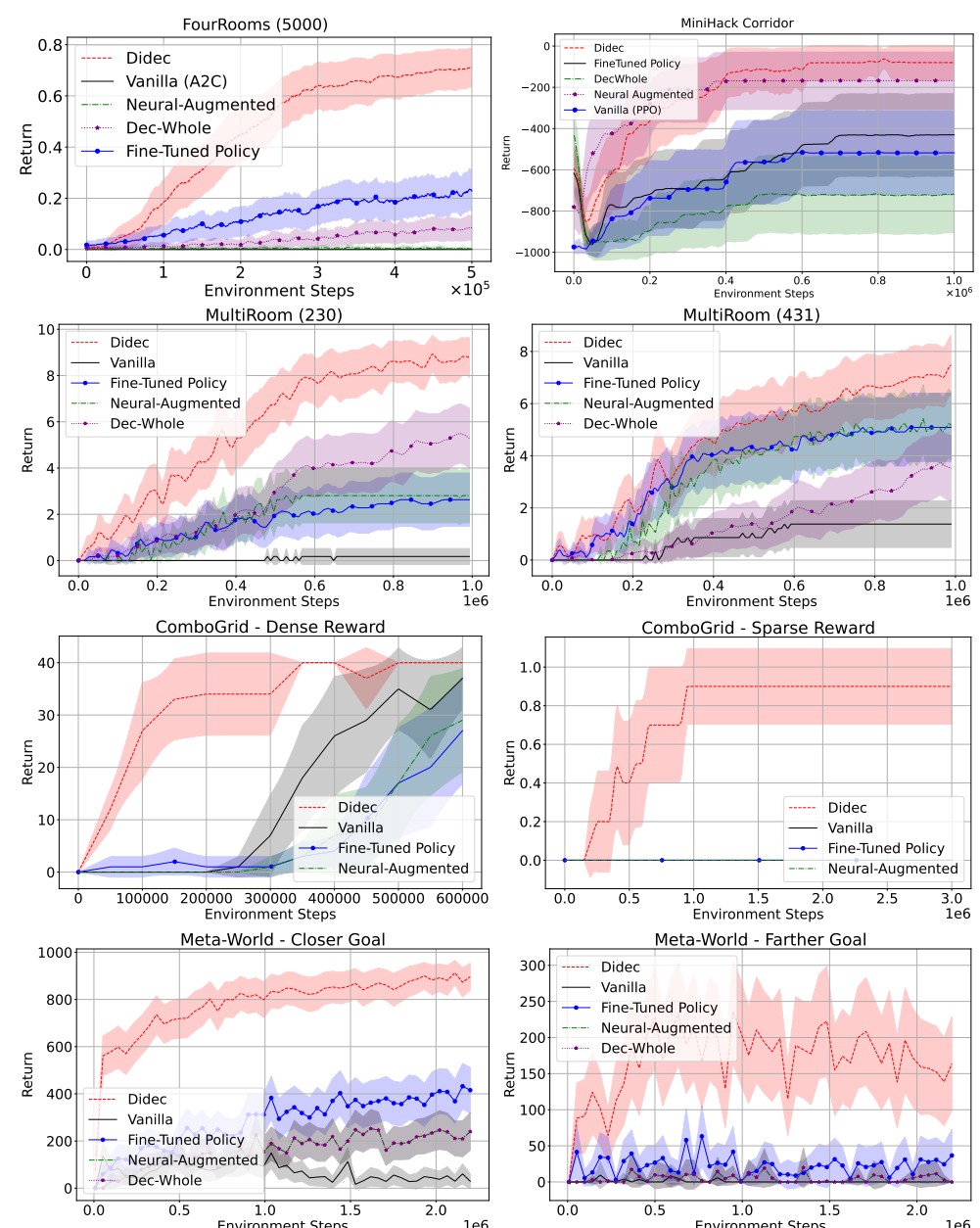

Figure 2: Learning curves of DIDEC and baselines: average return with 95% confidence intervals, using 50 seeds for FourRooms, 57 for the MultiRoom, 10 for ComboGrid, 22 for MiniHack, 97 for Meta-World.

discrete, we use a simplified version of DIDEC with Meta-World to handle its continuous spaces. When training masks, instead of considering all sub-trajectories of each $\mathcal{T}_j$ of $\mathcal{T}_{\text{train}}$, we consider only $\mathcal{T}_j$, thus resulting in one option per trajectory. Learned options are made available to the agent through a mixture-of-experts scheme (Jacobs et al., 1991) (Appendix L), thus bypassing the selection step. The experiments in Meta-World serve two purposes: to show that our mask-based decomposition can also be used in continuous spaces and in a simpler setting that bypasses the subset selection step.

Alikhasi & Lelis (2024) compared Dec-Options to a range of option-learning methods, including Modulating Masks (Ben-Iwhiwhu et al., 2022), Progressive Neural Networks (Rusu et al., 2016), Option-Critic (Bacon et al., 2017), ez-greedy (Dabney et al., 2021), and DCEO (Klissarov & Machado, 2023), and showed a clear advantage for Dec-Options. That advantage, shared by DIDEC, arises from

leveraging policies trained on $\{P_j\}_{j=1}^{i-1}$ to improve sample efficiency on $P_i$, which previous methods were not designed to leverage. To evaluate DIDEC, we compare it against baselines that also exploit previously learned policies. The first augments the action space with one-step options derived from earlier policies (**Neural-Augmented**). The second is like DIDEC except it reuses prior policies as a single undivided function rather than decomposing them (**Dec-Whole**). The third is also like DIDEC except that it fine-tunes the policy weights instead of training masks (**Fine-Tuned Policy**). We also use the base learning algorithm used with options (either A2C or PPO), but without options (**Vanilla**).

For each of the option-based approaches: DIDEC, Neural-Augmented, Dec-Whole, and Fine-Tuned, we augment the agent's action space with the set of options each method selects. Then, we train a policy on the test problem of each domain, an action-augmented agent with either A2C or PPO.

In Appendix H, we present the values considered in our hyperparameter sweep for DIDEC and baselines. We considered masking neurons, input features, or both, as hyperparameters of DIDEC. Our empirical results are presented in plots, where the y-axis represents the average return and the x-axis represents the number of agent interactions. In addition to the average value, we also present the 95% confidence intervals. The number of seeds used is shown in the caption of the plots.

## 7.1 EXPERIMENTAL RESULTS

Figure 2 presents the results. These results support our hypothesis that the gradient search DIDEC performs can extract sub-functions that help solve downstream tasks even when the space of sub-functions is too large to be tackled with complete enumeration.

The performance of DIDEC on MultiRoom (230) is more evident than on MultiRoom (431) and MiniHack. We conjecture that this happens because one of the rooms in problem 230 is much narrower than the rooms in the training problems (see Appendix I), which makes it more difficult to generalize. The structures encountered in the training of MultiRoom and MiniHack are more similar to those found in the test problem, thus allowing the baselines also to discover options that generalize. DIDEC also presents similar advantages over the tested baselines on the Meta-World tasks.

The results on ComboGrid (dense and sparse) demonstrate that DIDEC is also applicable to recurrent models. In this domain, Dec-Whole did not select any options, so it behaves like Vanilla. Each cell in Table 1 shows the fraction of DIDEC's (top) and Fine-Tuned Policy's (bottom) options that generalize to the unseen observations of the test problem; the gray cells denote goal locations, so we did not calculate their values. At least 60% of DIDEC's options, which encode helpful combos such as 'move, move', generalize, while Fine-Tune's options fail. Recall that Fine-Tune differs from DIDEC only in the weights it adapts—Fine-Tune adjusts model weights, whereas DIDEC learns masks. This result illustrates how extracting sub-functions with default parameters can enable generalization.

| **DIDEC** | | | | |
|---|---|---|---|---|
| 0.6 | 0.6 | | 0.6 | 0.6 |
| 0.6 | 0.6 | 0.6 | 0.6 | 0.6 |
| | 0.6 | 0.6 | 0.6 | |
| 0.6 | 0.6 | 0.6 | 1.0 | 0.6 |
| 0.6 | 0.6 | | 0.6 | 0.6 |
| **Fine-Tuned Policy** | | | | |
| 0.0 | 0.0 | | 0.0 | 0.0 |
| 0.0 | 0.0 | 0.2 | 0.0 | 0.0 |
| | 0.0 | 0.0 | 0.0 | |
| 0.0 | 0.0 | | 0.0 | 0.0 |
| 0.0 | 0.0 | | 0.0 | 0.0 |

Table 1: Generalization results of options generated by DIDEC and Fine-Tuned Policy in ComboGrid.

## 8 CONCLUSIONS

In this work we introduced Differentiable Dec-Options (DIDEC), a method for discovering reusable sub-functions of neural networks through gradient-based decomposition. Unlike the earlier enumerative approach Dec-Options, DIDEC scales to larger networks by casting sub-tree extraction as a differentiable masking problem. Beyond neuron masks, DIDEC also learns input masks that specify default parameters, enabling reuse of sub-functions in new contexts. We evaluated DIDEC across FourRooms, MultiRoom, MiniHack, and ComboGrid, showing that the extracted sub-functions improve sample efficiency on downstream tasks. More broadly, by treating network decomposition as a differentiable process, DIDEC allows for the construction of libraries of reusable programs from neural policies that would otherwise be too large to decompose. This is a step toward integrating the scalability of neural networks with the modularity and reuse of programmatic representations.

## REPRODUCIBILITY STATEMENT

The code used to run all experiments in this paper will be made available after the review process.

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

## A   GREEDY ALGORITHM AND LEVIN LOSS COMPUTATION

In this section, we explain the greedy algorithm Alikhasi & Lelis (2024) used to select subsets. In Dec-Options' greedy algorithm, we start with an empty $\Omega'$ and select the option $\omega$ from $\Omega$ that minimizes the sum of the Levin loss for the trajectories in $\Pi_{\text{train}}$ the most. Then, it makes $\Omega' = \{\omega\}$. In the next iteration, the procedure chooses another $\omega$ from $\Omega$ such that $\Omega' \cup \{\omega\}$ minimizes the sum of the Levin losses the most. This process continues until adding another option to $\Omega'$ would increase the Levin loss. The algorithm then stops and returns $\Omega'$ as its selected subset.

A key step in the greedy algorithm for subset selection is the computation of the Levin loss for a given subset $\Omega'$ and a trajectory $\mathcal{T}$. The algorithm 2 shows a dynamic programming approach to compute the loss. Such a procedure is necessary because the computation of the Levin loss depends on which options are used in each trajectory step. For example, if $\Omega' = \{\omega_1, \omega_2\}$ and both options can be applied in time step 1 of the trajectory, then for which should the loss be computed? Algorithm 2 shows an efficient procedure for considering all possibilities of use of options.

---

**Algorithm 2** COMPUTE-LOSS

**Require:** Sequence $\mathcal{S} = \{o_0, o_1, \cdots, o_{T+1}\}$ of states of a trajectory $\mathcal{T}$, probability $p_{u,\Omega}$, options $\Omega$
**Ensure:** $\mathcal{L}(\mathcal{T}, \pi_u^\Omega)$
1: ▷ *Initialize table P as if no options were available: to reach the j-th state we need j actions*
2: $M[j] \leftarrow j$ for $j = 0, 1, \cdots, T+1$
3: **for** $j = 0$ to $T+1$ **do**
4:    **if** $j > 0$ **then**
5:       $M[j] \leftarrow \min(M[j-1]+1, M[j])$
6:    **for** $\omega$ in $\Omega$ **do**
7:       **if** $\omega$ is applicable in $o_j$ **then**
8:          ▷ *Option $\omega$ is used in $o_j$ for $\omega_z$ steps*
9:          $M[j + \omega_z] \leftarrow \min(M[j+\omega_z], M[j]+1)$
10: ▷ $M[T+1]$ *stores the smallest number of actions to reach the end of the sequence. The value of $p_{u,\Omega}$ is the probability of taking an action in the option-augmented action space according to the uniform policy. The function returns the minimum Levin loss for $\mathcal{T}$ and $\Omega$.*
11: return $|\mathcal{T}| \cdot (p_{u,\Omega})^{-M[T+1]}$

---

## B   STOCHASTIC HILL CLIMBING FOR SUBSET SELECTION

Alikhasi & Lelis (2024) used a common approximation to the NP-hard subset selection problem (Garey & Johnson, 1979), which greedily and iteratively selects the option that minimizes the Levin loss the most (see Appendix A). Preliminary experiments favored a stochastic hill climbing (SHC) algorithm over the greedy approach for selecting a subset of options. We use SHC with both DIDEC and with baselines that require approximating a solution to the subset selection problem.

SHC is a hill-climbing approach with a stochastic neighborhood function. SHC starts with a randomly selected candidate solution $c$ and greedily selects the best neighbor $c'$ of $c$. If $c'$ has a better Levin loss value than $c$, the search continues with $c'$ as the new $c$. Otherwise, the search terminates and returns $c$. We use SHC with random restarts. Once a candidate is returned, we repeat it from another initial candidate. The SHC result is the candidate with the smallest loss across all restarts.

In DIDEC, a candidate solution is a subset $\Omega'$ of $\Omega$. To reduce the search space, all candidates considered in the search satisfy $|\Omega'| \leq s_{\max}$, where $s_{\max}$ is a hyperparameter that limits the maximum number of options that can be selected. The neighborhood function $\mathcal{N}$ is defined as follows. Given a candidate $\Omega'$, we sample $v$ options $\Omega''$ from $\Omega - \Omega'$. If $|\Omega'| < s_{\max}$, we generate $v$ neighbors $\Omega' \cup \{\omega\}$, one for each $\omega$ in $\Omega''$. We also generate $v^2$ neighbors, where each $\omega''$ in $\Omega''$ is used to replace an $\omega'$ in $\Omega'$. Finally, we generate other $|\Omega'|$ neighbors where we remove each $\omega'$ from $\Omega'$, thus generating neighbors of size $|\Omega'| - 1$. The function $\mathcal{N}$ returns the union of all these neighbors.

We sample the $v$ options from $\Omega''$ according to a distribution that favors options complementary to the current candidate subset $\Omega'$. Let $\mathcal{T}_{\text{train}}^{\Omega'}$ be the set of observation-action pairs from $\Pi_{\text{train}}$ that are

"not covered" by an option in $\Omega'$. Formally, the $j$-th observation-action pair of a trajectory is not covered by an option in $\Omega'$ if, while computing the Levin loss of $\Omega'$ in Algorithm 2, line 5, the $\min$ operator returns $M[j-1]+1$. Given the set of observation-actions pairs $\mathcal{T}_{\text{train}}^{\Omega'}$, we define the value of each $\omega$ in $\Omega''$ as the number of times $\omega$ can be initiated at a pair in $\mathcal{T}_{\text{train}}^{\Omega'}$. An option $\omega$ of the form `repeat(b):`$\pi_\omega$ can be initiated at a pair $(o_j, a_j)$ if $\pi_\omega(o_{j+i})$ returns $a_{j+i}$ for $i$ in $\{0, 1, \cdots, b-1\}$. We sample options from $\Omega''$ proportionally to their value. This neighborhood function favors options that can be used more often in pairs that are not yet covered by the options in the current candidate.

## C   DIDEC - OVERALL APPROACH

---

**Algorithm 3** Differentiable Dec-Options (DIDEC)

---

**Require:** Observation-action trajectories $\mathcal{T}_{\text{train}} = \{\mathcal{T}_j\}_{j=1}^{i-1}$, neural policies $\Pi_{\text{train}} = \{\pi_j\}_{j=1}^{i-1}$ that generated the trajectories in $\mathcal{T}_{\text{train}}$, maximum subset size $s_{\max}$, maximum length of an option $z_{\max}$, neighborhood function $\mathcal{N}_v$, number of epochs $E$, learning rate $\alpha$, a number of restarts $r$.

**Ensure:** A set of at most $s_{\max}$ options of the form `repeat(b):`$\omega_\pi$.

1:  ▷ *Generating training sequences by sliding a window of size $z = 2, \cdots, z_{\max}$ over the observation-action trajectories the policies in $\Pi_{train}$ generate.*
2:  $\mathcal{D} \leftarrow \emptyset, \Omega \leftarrow \emptyset$
3:  **for** $z = 2, 3, \ldots, z_{\max}$ **do**
4:      **for** each trajectory $\mathcal{T}_j$ in $\mathcal{T}_{\text{train}}$ **do**
5:          **for** $t = 0, 1, \ldots, T_j - z$ **do**
6:              $\mathcal{D} \leftarrow \mathcal{D} \cup \{(o_t, a_t), (o_{t+1}, a_{t+1}), \ldots, (o_{t+z-1}, a_{t+z-1})\}$
7:  ▷ *Training masks for selecting sub-trees of the neural trees of policies in $\Pi_{train}$. The masks can be input-only, neurons-only, or input-and-neurons. Each masked policy $\pi^\Theta$ results in an option.*
8:  **for** each sub-trajectory $\tau$ in $\mathcal{D}$ **do**
9:      **for** each policy $\pi$ in $\Pi_{\text{train}}$ **do**
10:         **if** $\pi$ generated $\tau$ **then**
11:             continue
12:         Initialize $\Theta$ randomly
13:         **for** epoch $= 1, 2, \ldots, E$ **do**
14:             $\hat{a} = \pi^\Theta(\tau)$, where $\pi^\Theta$ is $\pi$ parameterized by mask $\Theta$
15:             $\mathcal{L}(\Theta) = -\tau_a^\top \log(\hat{a})$
16:             $\nabla\Theta \leftarrow \frac{\partial \mathcal{L}(\Theta)}{\partial \Theta}$
17:             $\Theta \leftarrow \Theta - \alpha \cdot \nabla\Theta$
18:     $\Omega \leftarrow \Omega \cup \{\texttt{repeat}(|\tau|)\,\pi^\Theta\}$
19: ▷ *Perform hill climbing to select a subset of $\Omega$. The neighborhood function $\mathcal{N}_v(c, s_{\max})$ returns a set of neighbors after sampling $v$ options from the options not in $c$; each neighbor has at most $s_{\max}$ options, as described in Appendix B. `Compute-Loss` is as described in Algorithm 2.*
20: $c^* \leftarrow \emptyset, l^* \leftarrow \infty$
21: **for** restart $= 1, 2, \ldots, r$ **do**
22:     $c \leftarrow$ Random subset of $\Omega$ with size at most $s_{\max}$
23:     $l \leftarrow$ `Compute-Loss`$(c)$
24:     $c_{\text{best}} \leftarrow c, l_{\text{best}} \leftarrow l$
25:     **while** True **do**
26:         $i \leftarrow$ False
27:         **for** each neighbor $c' \in \mathcal{N}_v(c_{\text{best}}, s_{\max})$ **do**
28:             $l' \leftarrow$ `Compute-Loss`$(c')$
29:             **if** $l' < l_{\text{best}}$ **then**
30:                 $c_{\text{best}} \leftarrow c', l_{\text{best}} \leftarrow l', i \leftarrow$ True
31:         **if** not $i$ **then**
32:             break
33:     **if** $l_{\text{best}} < l^*$ **then**
34:         $c^* \leftarrow c_{\text{best}}, l^* \leftarrow l_{\text{best}}$
35: **return** The set of options $c^*$ represents

---

# D    LEVIN LOSS MINIMIZATION FOR OPTION SELECTION IS HARD

In this section, we prove that selecting a subset of options that minimizes the Levin loss is NP-hard.

**Definition 1** (The LLSS Problem). *Let $\mathcal{T} = \{\mathcal{T}_1, \ldots, \mathcal{T}_m\}$ be a set of trajectories and $\Omega = \{\omega_1, \ldots, \omega_n\}$ be a set of options. The* Levin loss subset selection *(LLSS) problem asks to find a subset $\Omega' \subseteq \Omega$ that minimizes $\sum_{i=1}^m \mathcal{L}(\mathcal{T}_i, \pi_u^{\Omega'})$, where $\mathcal{L}(\mathcal{T}_i, \pi_u^{\Omega'})$ is the Levin loss on trajectory $\mathcal{T}_i$ under the uniform policy $\pi_u^{\Omega'}$ over the primitive action set augmented with $\Omega'$.*

**Theorem 1.** *The LLSS problem is NP-hard.*

*Proof.* We reduce from the NP-hard Minimum Cover (MC) problem (Garey & Johnson, 1979, Problem SP5). In the MC problem, we are given a universe $U = \{u_1, \ldots, u_m\}$ and a collection of subsets $S = \{S_1, \ldots, S_n\}$. The goal is to find the smallest subcollection of $S$ whose union is $U$.

We create one trajectory $\mathcal{T}_i$ for each element $u_i \in U$; all trajectories have length 2. Each subset $S_j$ is mapped to an option $\omega_j$, giving $\Omega = \{\omega_1, \ldots, \omega_n\}$. Every option runs for 2 steps and is applicable on the first observation of $\mathcal{T}_i$ if and only if $u_i$ in $S_j$. Thus an option either covers a trajectory or does not. We construct $\Omega$ so that every trajectory is covered by at least one option.

For a given trajectory $\mathcal{T}_i$ and option subset $\Omega'$, the Levin loss is $\mathcal{L}(\mathcal{T}_i, \Omega') = 2(|A| + |\Omega'|)^{M_{\mathcal{T}_i, \Omega'}}$, where $|A|$ is the number of primitive actions and $M_{\mathcal{T}_i, \Omega'}$ is the minimum number of actions or option calls needed to generate $\mathcal{T}_i$ using $\Omega'$ (as computed by Algorithm 2). If $\mathcal{T}_i$ is covered by $\Omega'$, a single option call produces the entire trajectory, so $M_{\mathcal{T}_i, \Omega'} = 1$ and $\mathcal{L}(\mathcal{T}_i, \Omega') = 2(|A| + |\Omega'|)$. If it is uncovered, we must execute two primitive actions, so $M_{\mathcal{T}_i, \Omega'} = 2$ and $\mathcal{L}(\mathcal{T}_i, \Omega') = 2(|A| + |\Omega'|)^2$.

Let $u$ denote the number of uncovered trajectories. Then

$$\sum_{i=1}^m \mathcal{L}(\mathcal{T}_i, \pi_u^{\Omega'}) = 2\Big[(m - u)(|A| + |\Omega'|) + u(|A| + |\Omega'|)^2\Big]$$
$$= 2\Big[m(|A| + |\Omega'|) + u\big((|A| + |\Omega'|)^2 - (|A| + |\Omega'|)\big)\Big].$$

If all trajectories are covered ($u = 0$), the loss simplifies to $2m(|A| + |\Omega'|)$, which is minimized by selecting a cover of minimum size, exactly as in the MC problem. Thus, among fully covering subsets, LLSS and MC coincide. We now show that, in our LLSS instance, no subset $\Omega'$ with $u > 0$ can have smaller total Levin loss than any fully covering subset.

Let $\Omega''$ be any subset that covers all trajectories ($u = 0$). For every $\Omega'$ with $u > 0$, we require

$$2\Big[m(|A| + |\Omega'|) + u\big((|A| + |\Omega'|)^2 - (|A| + |\Omega'|)\big)\Big] > 2m(|A| + |\Omega''|). \tag{2}$$

Since $u > 0$, the left-hand side of Equation 2 is a quadratic polynomial in $|A|$ with positive coefficient $2u$, while the right-hand side is linear in $|A|$. All coefficients depend polynomially on $m$ and $n$, and $|\Omega'|, |\Omega''| \leq n$. Therefore, there exists a polynomially bounded threshold $A^\star = \text{poly}(m, n)$ such that for all $|A| \geq A^\star$, Equation 2 holds for every $\Omega'$ with $u > 0$ and every fully covering $\Omega''$.

With such a choice of $|A|$, any subset leaving even a trajectory uncovered has larger loss than every fully covering subset. Thus, every optimal solution to LLSS must cover all trajectories and must minimize $|\Omega'|$. Thus an optimal solution to LLSS solves MC. $\square$

# E    EXAMPLES OF LEARNED OPTIONS IN COMBOGRID

Figure 3 shows the options learned in ComboGrid. In this domain, the options learned are modular and encode well-defined behaviors encoded in the underlying LTSM, such as moving forward twice and turning right. The agent learned options that are independent of the observation, which is helpful in this domain, given that the combo of actions does not change based on the observation. The agent is able to solve the test problem by using only options.

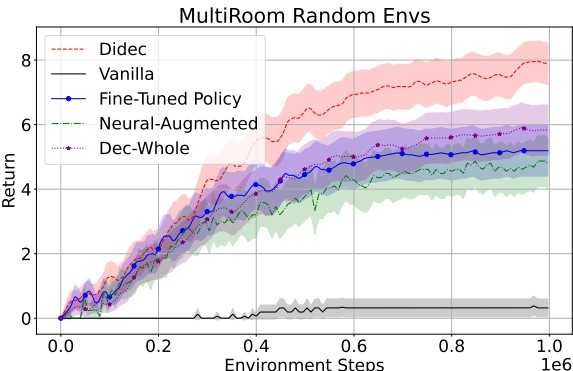

Figure 3: Sequence of actions DIDEC's agent learned on ComboGrid. Labels at the top present the semantics of a combo, while the labels at the bottom present the options learned.

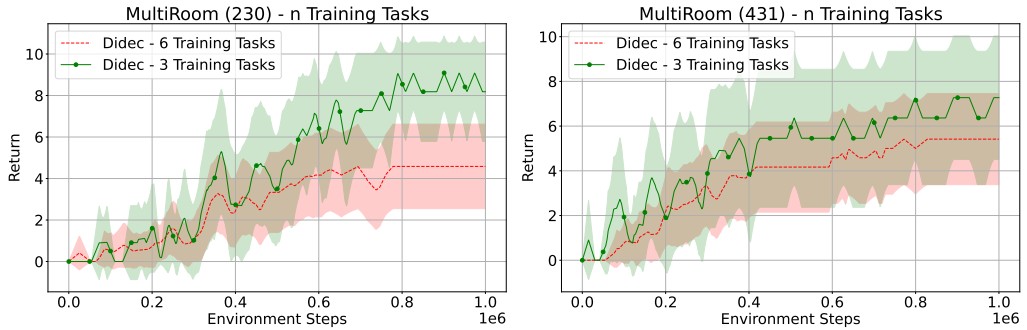

Figure 4: Learning curves of different systems evaluated when the training problems are selected randomly. The evaluation is performed on problems 230 and 431.

## F  VARYING THE TRAINING ENVIRONMENTS

The quality of the options DIDEC learns depends on the problems $\{P_j\}_{j=1}^{i-1}$ used for training. For example, in ComboGrid, the agent needs to learn a given combo in at least two problems to be able to extract an option encoding such a combo. To illustrate, let $\pi_1$ and $\pi_2$ be the policies encoding the combo to move forward. The masking process will then attempt to extract a sub-function of $\pi_1$ so that it can be applied to the problem $\pi_2$ solves. While it is clear what we need for the training problems of ComboGrid, it is not clear what is needed for the other domains evaluated. The selection of the training problems used in the other domains was arbitrary. To further explore this, we performed experiments on MultiRoom, where three training problems were selected randomly. The results are presented in Figure 4, where the learning curves are on the test problems with seeds 230 and 431. DIDEC's sample efficiency advantage over the baselines is similar to what we observed in Figure 2.

Figure 5: DIDEC's performance decreases when increasing from three to six training problems.

We also evaluate whether DIDEC is robust to increasing the number of training problems. Naturally, increasing the training set will increase the running time, as detailed in Section 5.2; here we evaluate the impact of a larger training set on the agent's sample efficiency. We compared an agent that uses DIDEC's options learned from three and six training problems in MultiRoom. The results are presented in Figure 5. We notice a decrease in performance as we increase the number of training tasks. Since the smaller training set is a subset of the larger one, the decrease in performance could be explained by the hill-climbing search being unable to find helpful subsets of options in the larger $\Omega$ set. This issue can potentially be mitigated by increasing the search budget in the selection step.

## G  NEURAL ARCHITECTURES

We adopt a neural Actor-Critic framework. The model is composed of two networks with shared input: an actor that parameterized a stochastic policy, and a critic that estimates the value function.

Let $d_{\text{obs}}$ denote the dimensionality of the observation vector (after flattening), and let $|\mathcal{A}|$ be the number of discrete actions.

### G.1  FEEDFORWARD ARCHITECTURE

The following network was used in FourRooms and MultiRoom. The architecture used with MiniHack is identical, except that it uses 256 neurons instead of 64.

**Actor Network.**  The actor network maps an input observation to a distribution over actions via the following architecture:

- A linear layer with input size $d_{\text{obs}}$ and output size 64,
- ReLU activation,
- A final linear layer with output size $|\mathcal{A}|$ producing action logits.

The final layer is initialized with a reduced standard deviation (std $= 0.01$) to promote stability during early exploration.

**Critic Network.**  The critic network has a deeper architecture to estimate the state value:

- A linear layer from $d_{\text{obs}}$ to 64 units,
- ReLU activation,
- Another linear layer with 64 hidden units,
- ReLU activation,
- A final linear layer mapping to a scalar value.

**Weight Initialization.**  Weights are initialized orthogonally with a gain factor of $\sqrt{2}$, and biases are set to $0.0$). The final layer of the critic is initialized with standard deviation of $1.0$.

### G.2  LSTM ARCHITECTURE

We use the Stable-Baselines3 (SB3) (Raffin et al., 2021) implementation of PPO with an LSTM representation. The policy consists of: (i) two unidirectional LSTMs, one for the actor and another for the critic, with 16 neurons that produce a latent state; and (ii) separate, feedforward network heads for actor and critic.

**Actor Network.**  From the last LSTM output at each timestep, the actor head applies:

- A linear layer with 32 units,
- $\tanh$ activation,
- A final linear layer with output size $|\mathcal{A}|$ producing action logits.

| Hyperparameter | Values Tested |
|---|---|
| Actor Learning rate | $\{0.01, 0.001, 0.0001\}$ |
| Critic Learning rate | $\{0.01, 0.001, 0.0001\}$ |
| Training time steps | $\{1\ \text{million}\}$ |
| Rollout depth | $\{5, 10, 20\}$ |
| Seeds per config | $\{0, 1, 3\}$ |

Table 2: Hyperparameters evaluated for DIDEC on FourRooms (5000).

| Methods | Actor learning rate | Critic learning rate | Rollout depth |
|---|---|---|---|
| Didec | 0.001 | 0.001 | 20 |
| Fine-Tune | 0.001 | 0.01 | 20 |
| Dec-Whole | 0.001 | 0.0001 | 20 |
| Neural-Augmented | 0.0001 | 0.001 | 5 |
| Vanilla | 0.001 | 0.001 | 20 |

Table 3: Chosen hyperparameters per method on FourRooms (5000).

**Critic Network.** The critic shares the same LSTM core but uses its own head:

- A linear layer with 32 units,
- tanh activation,
- A final linear layer mapping to a scalar value.

**Weight Initialization.** Weights are initialized orthogonally with a gain factor of $\sqrt{2}$, and biases are set to $0.0$. The final actor layer is initialized with a smaller weight scale of $0.01$. The final critic layer is initialized with a gain of $1.0$.

## H  HYPERPARAMETER SWEEPS

The hyperparameter sweeps were performed with a grid search over the values shown in the tables of this section. We considered three seeds for each set of values, and the value of a set is the average area under the curve (AUC) of the agent's return across the three seeds. We report the result of each approach using the hyperparameter set that has the largest average AUC.

Tables 2 and 3 present the sweep results for FourRooms, while Tables 4, 5, and 6 present the sweep results for MultiRoom. In some cases, the value selected in the sweep is a "border value", i.e., the smallest or the largest value attempted in the sweep. While methods using border values could perform better by increasing the size of our sweep, this experimental bias is present in all evaluations, not just the baselines. This means that DIDEC's performance could also be improved in some cases.

### H.1  SELECTING MASKING TYPE

We considered masking neurons, input features, or both as a hyperparameter. The results presented in the paper are based on the selection of the best-performing scheme for DIDEC. Figure 6 presents the performance of DIDEC when masking only the input features, only the neurons, and both. While there is no noticeable difference among the different MultiRoom environments, masking input features significantly improves performance in FourRooms. Note that, by choosing default values for the inputs, we select sub-trees of the underlying neural tree, implicitly masking neurons. An advantage of masking neurons, which we have not explored in this work, is that it works as a compression scheme. Every neuron that is set to either active or inactive can be removed from the network, thus compressing the model. Another advantage of neuron masking that we have not explored is the ability to mask neurons when the input space is high-dimensional, such as in pixel inputs. In this type of input, masking neurons might represent an easier optimization problem than masking inputs.

| Hyperparameter | Values Tested |
|---|---|
| Learning rate | $\{0.01, 0.005, 0.001, 0.0005, 0.00005\}$ |
| Clipping coefficient | $\{0.01, 0.05, 0.1, 0.15, 0.2, 0.3\}$ |
| Entropy coefficient | $\{0.0, 0.01, 0.02, 0.03, 0.05, 0.1, 0.2\}$ |
| Training time steps | $\{1\ \text{million}\}$ |

Table 4: Hyperparameters evaluated for DIDEC on MultiRoom.

| Methods | Learning rate | Clipping coefficient | Entropy coefficient |
|---|---|---|---|
| Didec | 0.01 | 0.2 | 0.01 |
| Fine-Tune | 0.01 | 0.3 | 0.02 |
| Dec-Whole | 0.001 | 0.3 | 0.02 |
| Neural-Augmented | 0.01 | 0.15 | 0.02 |
| Vanilla | 0.001 | 0.3 | 0.01 |

Table 5: Chosen hyperparameters per method on MultiRoom (230).

Figure 7 shows the DIDEC results for LSTM policies on ComboGrid (dense), where one uses options learned only by learning the hidden state and memory vectors of the LSTM (LSTM Only), while the other learns the initial hidden state and memory vectors in addition to input masking (input and LSTM). These results highlight the need for both selecting the initial vectors and masking the inputs. A helpful analogy is that by learning the initial vectors, we select the initial mode of the LSTM's underlying FSM; by masking the inputs, we control which transitions we allow from that initial mode. By doing both, we are selecting a sub-automaton of the LSTM underlying FSM.

# I   PROBLEM DOMAINS

This section shows the environments used in our experiments. Figure 8 shows the ten SimpleCrossing environments used in training, and the FourRooms environments used in testing. Figure 9 shows the training and test problems for MultiRoom. Interestingly, DIDEC outperforms the baselines by a large margin on problem 230 (Figure 2), whose middle room has a structure different than the training problems—the room is narrower than the rooms in problems 1, 3, and 17. This discrepancy can affect generalization because the agent has not previously observed this type of room. This result demonstrates the generalization capabilities of the learned options. For both FourRooms and MultiRoom, we use MiniGrid's standard reward function (Chevalier-Boisvert et al., 2023).

Figure 10 shows the four ComboGrid training problems and the test problem, where the agent needs to collect all four markers to complete the task. In the dense version of the environment, each marker gives a reward of ten; any other step gives zero reward. In the sparse version, the agent receives one after collecting all markers, and zero otherwise.

Figure 11 shows the training and test problems of the Corridors environment we used from MiniHack. In this environment, the agent receives a reward of -1 for every step, except for the transition to the stairs, when the episode terminates.

# J   EXPLORATION WITH DIDEC'S OPTIONS

Figure 13 presents the heatmaps denoting the probability of an agent visiting each cell in the first 500,000 training steps of the A2C agent, for Vanilla, Dec-Whole, Fine-Tune and DIDEC (from left to right). DIDEC's allows for better exploration as the agent learns to navigate between rooms much more quickly than Vanilla, as evidenced by the lighter rectangle connecting the four rooms.

| Methods | Learning rate | Clipping coefficient | Entropy coefficient |
|---|---|---|---|
| Didec | 0.001 | 0.2 | 0.03 |
| Fine-Tune | 0.01 | 0.2 | 0.02 |
| Dec-Whole | 0.001 | 0.3 | 0.01 |
| Neural-Augmented | 0.005 | 0.15 | 0.05 |
| Vanilla | 0.001 | 0.3 | 0.01 |

Table 6: Chosen hyperparameters per method on MultiRoom (431).

| Hyperparameter | Values Tested |
|---|---|
| Learning rate | $\{0.01, 0.005, 0.001, 0.0001\}$ |
| Clipping coefficient | $\{0.1, 0.2, 0.3\}$ |
| Entropy coefficient | $\{0.0, 0.01, 0.02, 0.05\}$ |
| Rollout Length | $\{128\}$ |
| Training time steps | $\{1 \text{ million}\}$ |

Table 7: Hyperparameters evaluated for DIDEC on ComboGrid - Sparse Reward.

## K    TIME ANALYSIS OF OPTION EXTRACTION

To quantify the computational overhead introduced by option extraction and option selection, we measured wall-clock time for each system evaluated on MiniGrid SimpleCrossing and our Meta-World task under an identical compute setup. For MiniGrid, all methods run on a single CPU job with 16 physical CPU cores and 32 parallel worker processes (Python multiprocessing). For Meta-World we use a single CPU with a single process. No GPUs are used in these experiments.

For DIDEC, FineTune, and DecWhole we report the average and standard deviation of

1. Learning Time: time to learn all candidate options across the base problems.

2. Selection Time: time to select a subset that reduces the Levin loss.

3. Total Time: the sum of the two steps.

All results shown in Table 15. In MiniGrid, DIDEC's running time is comparable to that of Fine-Tune, but with a much improved sample efficiency (see Figure 2). DIDEC's running time of learning options is on par with the time the agent spends exploring the environment. This is loosely related to wake-sleep algorithms (Hinton et al., 1995). In DIDEC, the agent goes through its wake cycle, where it learns to solve problems and its sleep cycle it organizes what it has learned into reusable options. The experiments in Meta-World shows that DIDEC's neural decomposition can be used in much simpler settings, where the running time can be reduced from minutes to seconds.

## L    DIDEC WITH CONTINUOUS ACTION SPACES

In our Meta-World experiments, we use a mixture-of-experts architecture (Jacobs et al., 1991), combining the base policy's action with that of one or more options. We use Stable-Baselines3 PPO with a Gaussian policy (Raffin et al., 2021). The policy outputs state-dependent means and log standard deviations, and actions are sampled using the reparameterization trick:

$$a = \mu(o) + \sigma(o) \odot \epsilon, \qquad \epsilon \sim \mathcal{N}(0, I),$$

where $\mu(o)$ and $\sigma(o)$ are the outputs of the policy network for observation $o$, and $\odot$ denotes element-wise multiplication. Given the base action $a$ and the option action $a_\omega$, the final action is

$$a_{\text{mix}} = \theta_1 a + \theta_2 a_\omega, \qquad \theta_1, \theta_2 \geq 0, \quad \theta_1 + \theta_2 = 1,$$

| Hyperparameter | Values Tested |
|---|---|
| Learning rate | $\{0.005, 0.001, 0.0001, 0.0005\}$ |
| Clipping coefficient | $\{0.1, 0.2, 0.3\}$ |
| Entropy coefficient | $\{0.0, 0.01, 0.02, 0.05\}$ |
| Rollout Length | $\{128, 256\}$ |
| Training time steps | $\{3\ \text{million}\}$ |

Table 8: Hyperparameters evaluated for DIDEC on ComboGrid - Dense Reward.

| Methods | Learning rate | Clipping coefficient | Entropy coefficient |
|---|---|---|---|
| Didec | 0.005 | 0.3 | 0.01 |
| Fine-Tune | 0.005 | 0.1 | 0.01 |
| Neural-Augmented | 0.005 | 0.1 | 0.01 |
| Vanilla | 0.005 | 0.1 | 0.02 |

Table 9: Chosen hyperparameters per method on ComboGrid - Dense Reward. The chosen rollout length for all the methods was 128.

where the mixture weights $\theta$ are produced by a softmax gating network and learned end-to-end through PPO's gradients. This extends naturally to $K$ options by assigning one weight $\theta_i$ per option

$$a_{\text{mix}} = \sum_{i=1}^{K} \theta_i a_i, \qquad \theta_i \geq 0, \quad \sum_{i=1}^{K} \theta_i = 1.$$

Unlike the classical options framework (Sutton et al., 1999), the actions produced by an option $\omega$ are not temporally extended. Instead, each option provides a state-conditioned continuous action at every timestep. This formulation demonstrates that DIDEC's mask-based decomposition is useful beyond option execution and can be combined with standard RL policies in continuous domains.

## M  LARGE LANGUAGE MODEL USAGE

We used LLMs to edit the text, particularly to trim words and correct ungrammatical sentences. We also used LLMs as a coding assistant in the development of the algorithms evaluated in this paper.

| Methods | Learning rate | Clipping coefficient | Entropy coefficient |
|---|---|---|---|
| Didec | 0.001 | 0.1 | 0.01 |
| Fine-Tune | 0.001 | 0.2 | 0.02 |
| Neural-Augmented | 0.001 | 0.2 | 0.02 |
| Vanilla | 0.001 | 0.2 | 0.02 |

Table 10: Chosen hyperparameters per method on ComboGrid - Sparse Reward.

| Hyperparameter | Values Tested |
|---|---|
| Actor Learning rate | $\{0.0001, 0.0003, 0.0005\}$ |
| Critic Learning rate | $\{0.0001, 0.0003, 0.0005\}$ |
| Clipping Coefficient | $\{0.1, 0.2\}$ |
| Entropy Coefficient | $\{0.0, 0.01, 0.02, 0.05\}$ |
| Training time steps | $\{1 \text{ million}\}$ |

Table 11: Hyperparameters evaluated for DIDEC on MiniHack (30).

| Methods | Actor learning rate | Critic learning rate | Entropy coefficient |
|---|---|---|---|
| Didec | 0.0001 | 0.0005 | 0.05 |
| Fine-Tune | 0.0005 | 0.0003 | 0.02 |
| Dec-Whole | 0.0001 | 0.0005 | 0.02 |
| Neural-Augmented | 0.0003 | 0.0005 | 0.02 |
| Vanilla | 0.0005 | 0.0005 | 0.02 |

Table 12: Chosen hyperparameters per method on MiniHack (30).

| Hyperparameter | Values Tested |
|---|---|
| Learning rate | $\{0.0008, 0.0003, 0.0001, 0.00005\}$ |
| Entropy coefficient | $\{0.0, 0.001, 0.005\}$ |
| Training time steps | $\{200000\}$ |

Table 13: Hyperparameters evaluated for baselines on Meta-World.

| Methods | Learning rate | Entropy coefficient |
|---|---|---|
| Didec | 0.0003 | 0.001 |
| Fine-Tune | 0.0003 | 0.0 |
| Dec-Whole | 0.00005 | 0.001 |
| Neural-Augmented | 0.00005 | 0.001 |
| Vanilla | 0.0001 | 0.005 |

Table 14: Chosen hyperparameters per method on Meta-World.

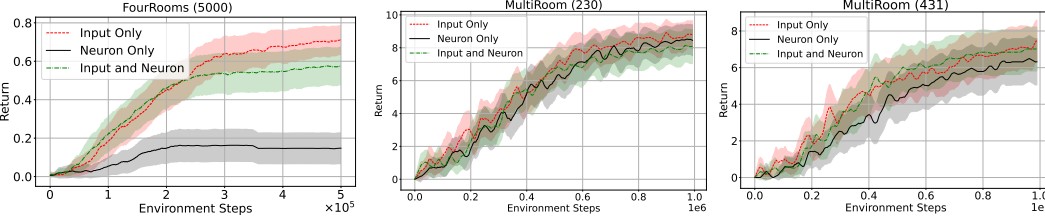

Figure 6: Evaluating masking schemes: input only, neurons only, and both. The plots present the average return and the 95% confidence intervals for 50 (FourRooms) and 57 seeds (MultiRoom).

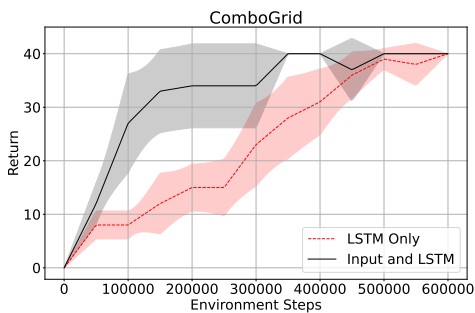

Figure 7: Comparison between learning only the initial hidden state and memory vectors of the LSTM (LSTM only), and learning the vectors and also masking the inputs.

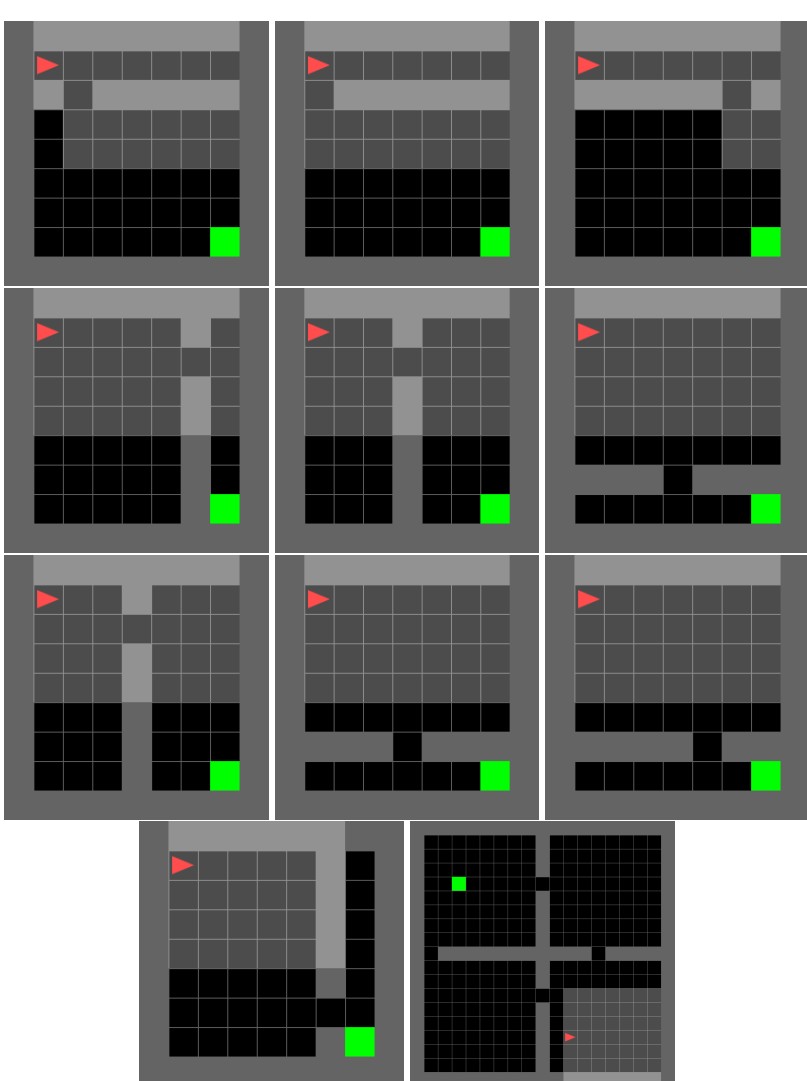

Figure 8: SimpleCrossing problems used as training for the FourRooms experiments. The seeds used to generate the problems, from left to right and top to bottom: 1000, 2000, 3000, 4000, 5000, 6000, 7000, 8000, 9000, 10000. The bottom-right image shows the FourRooms environment used (5000).

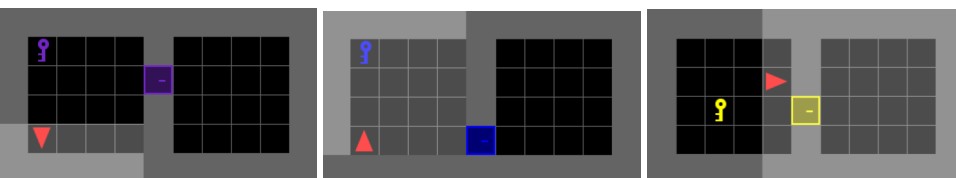

(a) Training problems used in MultiRoom. The seeds used to generate the problems, from left to right: 1, 3, 17.

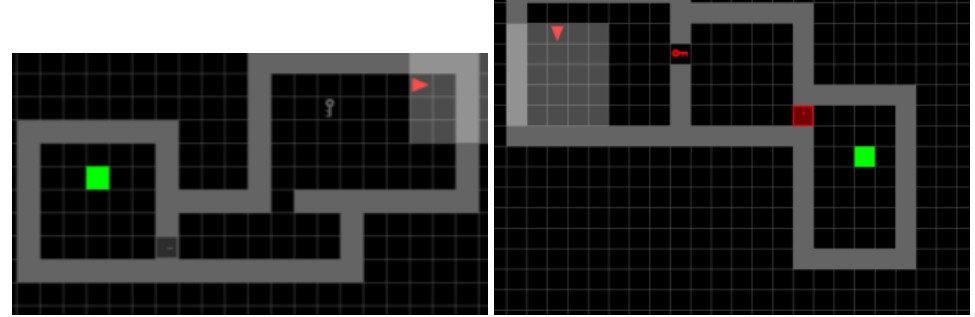

(b) Test problems used in MultiRoom. The seeds used to generate the problems, from left to right: 230 and 431.

Figure 9: Training and test problems used in our MultiRoom experiments.

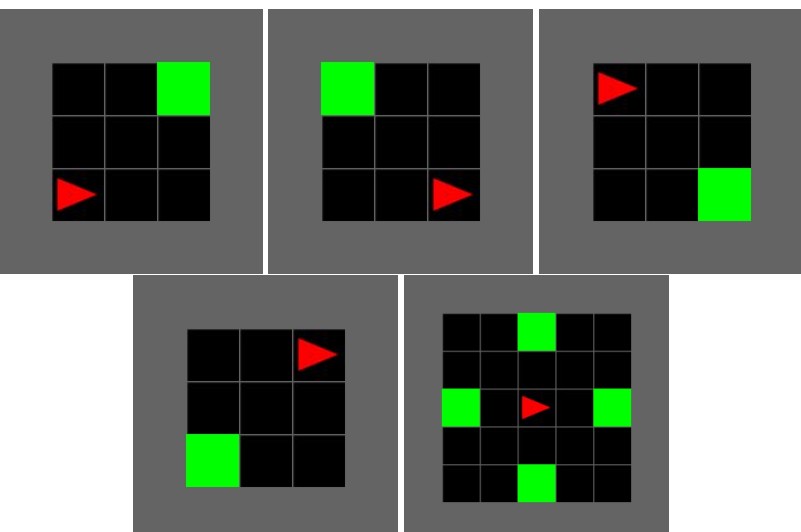

Figure 10: The four ComboGrid training environments, from left to right and top to bottom, and the test environment.

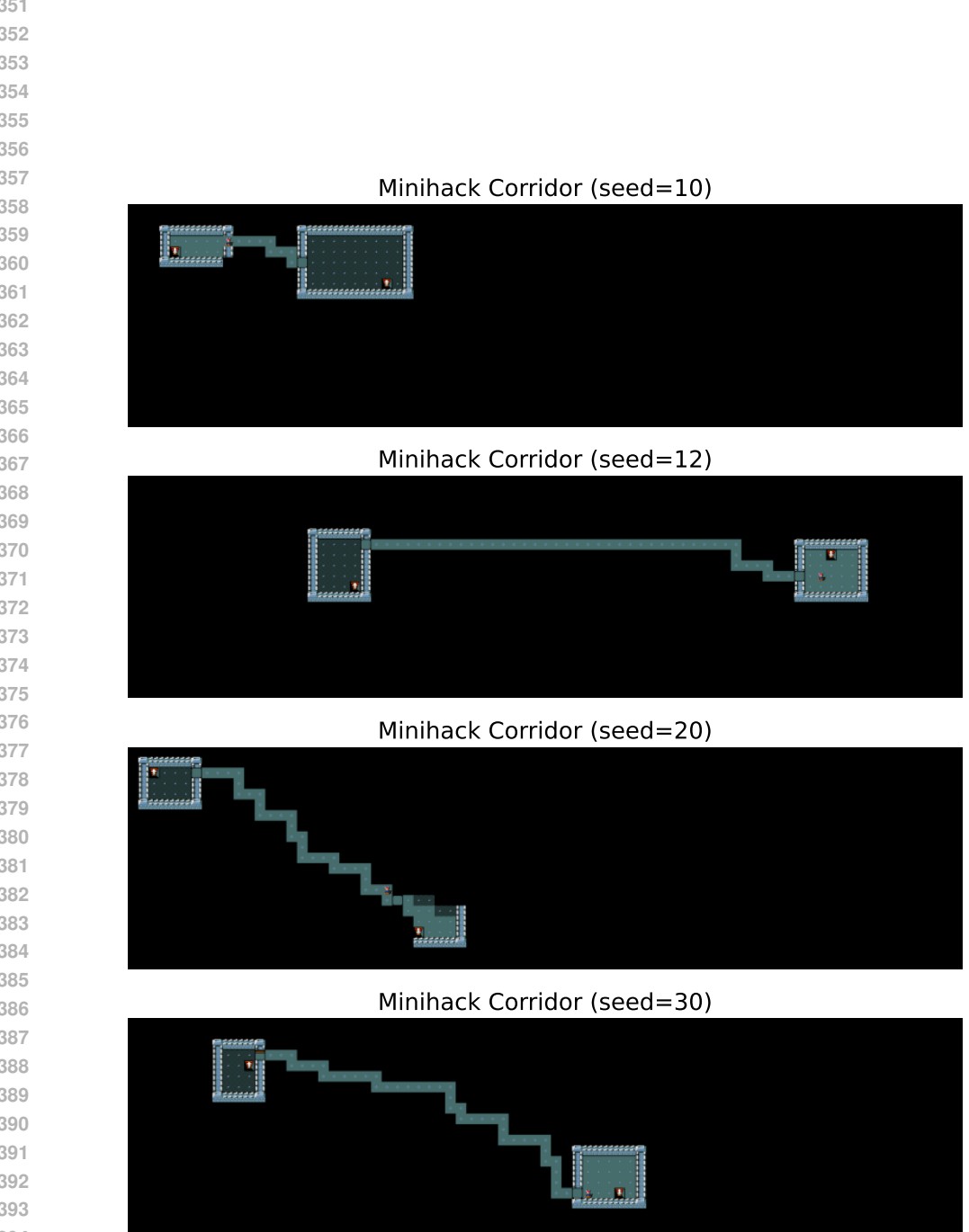

Figure 11: The three MiniHack Corridor training environments (top), and the test environment (bottom), generated with seeds 10, 12, 20, and 30, respectively.

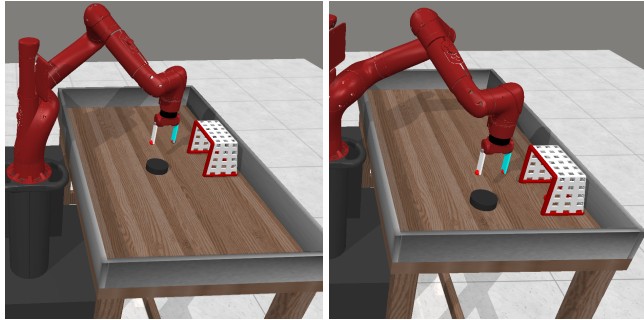

Figure 12: The two Meta-World training environments. In addition to the location of the net, the plate, and the arm, the distance between the plate and the net was changed in the test problems. The plate is closer to the goal in one test problem and farther in the other.

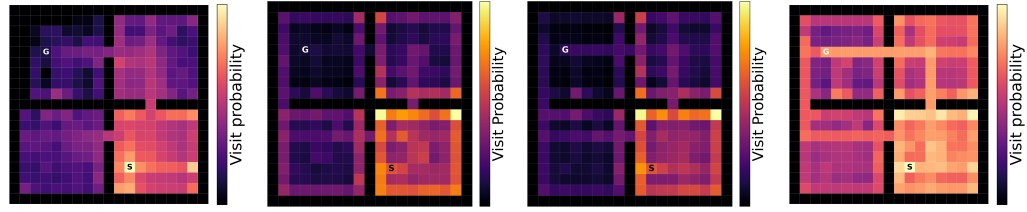

Figure 13: Probability of an agent visiting each of the cells in FourRooms in the first 500,000 training steps. From left to right Vanila, Dec-Whole, Fine-Tune, DIDEC. The data is averaged over 50 seeds.

| MiniGrid | | | |
|---|---|---|---|
| **Method** | **Learning** | **Selection** | **Total** |
| DIDEC | $1476.3 \pm 760.4$ | $1359.4 \pm 631.1$ | $2835.7 \pm 1390.3$ |
| Fine-Tune | $871.0 \pm 464.9$ | $1387.1 \pm 668.8$ | $2258.1 \pm 1131.0$ |
| Dec-Whole | $0.0 \pm 0.0$ | $53.4 \pm 17.8$ | $53.5 \pm 17.8$ |
| **Meta-World** | | | |
| **Method** | **Learning** | **Selection** | **Total** |
| DIDEC | $10.21 \pm 0.05$ | - | $10.21 \pm 0.05$ |
| Fine-Tune | $5.28 \pm 0.02$ | - | $5.28 \pm 0.02$ |
| Dec-Whole | $0.0 \pm 0.0$ | - | $0.0 \pm 0.0$ |

Table 15: Average running time in seconds and standard deviation for option learning and selection across different approaches. Dec-Whole does not perform any training, so the time for learning is zero. The Meta-World experiments do not include a selection step because we use one option per base policy. We use 50 seeds for MiniGrid and 40 for Meta-World.

