# OpenReview forum: "Option Discovery via Differentiable Neural Decomposition"
_ICLR.cc/2026/Conference — Submitted to ICLR 2026_

### Official Review · Reviewer_TdxZ · 2025-10-27

**Soundness:** 3
**Presentation:** 3
**Contribution:** 3
**Rating:** 8
**Confidence:** 3

**Summary:**

# Summary

The paper expands on a prior work by Alikhasi & Lelis 2024 to alleviate two limitations that prevented from applying that prior work in practical setting. The first one is the inability to consider a large portion of the combinatorially large space of sub-trees of so-called "neural trees", a result of decomposing neural networks into chains of if-then-else statements. The second one, the so-called "default input parameters" that encode values of features not present in a different observation from the one the policies were learned for, a must-have to make the policies generalize.

The work is based on a neat observation that masking neurons in the network can allow devising the corresponding sub-tree of the neural tree. Thus, both learning the sub-trees and the default parameters can be done with gradient descent by masking the input observation.


# Significance

This work makes previously suggested idea practically applicable, which makes it significant in my book.

# Soundness

The authors claim that "The process of selecting a subset of options that minimizes the Levin loss is NP-hard in general (Garey & Johnson, 1979)". I am not sure where in the cited book is the mentioned problem, it would be great to refer to it specifically (e.g., page number)


# Novelty

The novelty of the paper is somewhat lessens by this being a follow up work, but the paper does solve a real issue with the previously proposed approach, in a novel way.


# Scholarship

The scolarship seems to be fair, with the related work covering several related directions.

# Clarity

The paper is sufficiently clear, easy to follow, although sometimes feels like parts of the text were rephrased from previous versions by different people. However, I did not find it to be an issue.


# Evaluation and Reproducibility

The evaluation specifies precisely the hypotheses tested, and tests on a variety of environments, compared to other options learning methods (baselines). DiDec shows consistently high performance, compared to the baselines.


# Minor
1. line 073: the sentence is awkwardly composed, as if only part of it was changed into active/passive voice.
2. line 099: First reference to Figure 1 is on page 2, while the figure itself is on page 5. Consider bringing closer.

**Strengths:**

1. Neat observation that allows to make existing theoretical idea practically applicable.
2. Sound presentation, motivating example.
3. Extensive experimental evaluation.

**Weaknesses:**

1. The motivating example can be presented in a more structured way.
2. The complexity result is unclear, is it known from the literature (probably not)? If not, should be presented more formally.

**Questions:**

1. You say "The process of selecting a subset of options that minimizes the Levin loss is NP-hard in general (Garey & Johnson, 1979)". Is it really a known result? If not, do you have a proof for the claim? Even if it is, could you specify a corresponding optimization problem (e.g., integer program)? Integer programming solvers can deal with quite large problems, maybe a greedy approach is not actually needed?

2. I wonder whether there is a way to make your subtrees/options symbolically representable. Do you have a way to recognize what do the learned options correspond to in each of the tested domains? For instance, do the learned options policies in MiniGrid correspond to moving between rooms? Something else?

---

> ### Author Response · Authors · 2025-11-28
>
> **Comment:** You say "The process of selecting a subset of options that minimizes the Levin loss is NP-hard in general (Garey & Johnson, 1979)". Is it really a known result? If not, do you have a proof for the claim? Even if it is, could you specify a corresponding optimization problem (e.g., integer program)? Integer programming solvers can deal with quite large problems, maybe a greedy approach is not actually needed?
>
> **Answer:** Thank you for this question. Although minimizing the Levin loss is related to the subset selection problem, it is not identical to it. We have added a proof that selecting a subset of options that minimizes the Levin loss is NP-hard. We do so by reducing from the Minimum Cover problem (Problem SP5 in the book by Garey & Johnson). Please see Appendix D.
>
> **Comment:** I wonder whether there is a way to make your subtrees/options symbolically representable. Do you have a way to recognize what do the learned options correspond to in each of the tested domains? For instance, do the learned options policies in MiniGrid correspond to moving between rooms? Something else?
>
> **Answer:** Yes, we see the learned options as symbols that are extracted from neural networks. Some of these symbols are interpretable, as those learned for ComboGrid (see Appendix E). Some of them help with exploration, but are not necessarily interpretable, such as those learned for MiniGrid. Currently, we don’t have an automated process to recognize these symbols, but this question could lead to interesting research in explainable RL.

---

### Official Review · Reviewer_Nv3Y · 2025-10-28

**Soundness:** 3
**Presentation:** 1
**Contribution:** 2
**Rating:** 2
**Confidence:** 4

**Summary:**

This paper proposes Differentiable Dec-Options (DIDEC), a method for discovering reusable, temporally-extended "options" from previously trained neural network policies. The work's primary claim is that it overcomes the exponential combinatorial search problem of its predecessor, Dec-Options, which limited that method to impractically small networks. DIDEC reframes this combinatorial search as a differentiable masking problem, allowing it to scale to larger networks (e.g., 64-256 neurons). A secondary contribution is the concept of "default input parameters," learned via input masking, which allows extracted sub-functions to generalize to new states. The method is evaluated in discrete grid-world domains, where it shows improved sample efficiency over baselines.

**Strengths:**

1.  The core technical idea of reframing the $3^d$ combinatorial sub-tree search into a differentiable 3-way masking problem is clever.
2.  The insight of "default input parameters" to disentangle a skill from its context is a useful contribution to the sub-problem of skill generalization.
3.  The paper is well-written, with clear motivation and a strong illustrative example in Figure 1.

**Weaknesses:**

1.  Misleading Contribution (BC vs. RL): The paper's primary flaw is framing itself as an "option discovery" paper. The algorithm is a supervised, brute-force behavior cloning system that clones every possible sub-trajectory of length $z \in [2, z_{max}]$. This has little to do with the elegant goal-driven formulation of the options framework with the underlying option induced SMDP.
2.  Impractical Computational Complexity: The paper claims to solve the scalability problem of Dec-Options but ignores the $O(N \times T \times z_{max} \times E \times C_{train})$ complexity of its own generation step. This is not a "scalable" solution; I ackowledge it's improvements over Dec-Option, but as a work aims to scale this direction up, current complexity of DIDEC is doubtful to be applied to actual RL envs.
3.  Severely Limited Scope and Insufficient Validation: Validation on grid-worlds alone is not sufficient for ICLR in 2025. If this is a theory paper its acceptable, but as stated below, this work clearly lack of theoretical analysis. Standard benchmarks (MuJoCo, DMC, robotics) that define modern RL research was not tested at all.
4.  Stacked Heuristics: The method is not a principled solution. It relies on a proxy loss (cross-entropy) to generate options and a separate, non-optimal heuristic search (SHC) to select them. This stack of approximations lacks any formal guarantees. How is the convergence of each sub-option is guranteed (since sub-trajectory are segmented brute-forcely by sliding window, trained on trajectory not even sampled from its own)?  How is the optimality guaranteed?
5.  Authors need to work on the concise and accurate math derivations. Most parts of the algo are verbally explained and very hard to follow. It would be much ease for readers if they explain with equations / pseudo code.

**Questions:**

I acknowledge the improvements over Dec-Option of this work but raised my concerns as in weakness section. I do think the direction Dec-Option proposed is an exciting and novel direction worth to explore. But the claim authors made at the beginning "scaling Dec-Option up" turns to be a disappointment. I'm willing to raise my score if the author can prove their scalability to "real" problem such as locomotion or VLA tasks.

---

> ### Author Response · Authors · 2025-11-28
>
> **Comment:** Misleading Contribution (BC vs. RL).
>
> **Answer:** We agree that Didec's approach to learning options is based on behavior cloning. We have updated Section 5.1.1 to state this. However, we respectfully disagree that the contribution is misleading for using the options framework. The options Didec discovers are well-defined according to the original work of Sutton et al. (1999) (see Section 4).
>
> **Comment:** Impractical Computational Complexity.
>
> **Answer:** The complexity of Didec is polynomial in the input size, and thus practical. The correct time complexity for learning options is  $O\left(T_{\max}\,|\mathcal{T}_{\text{train}}|^{2}\right)$. This is because $z$ (maximum sub-trajectory length), $E$ (training epochs), and $P$ (number of parameters of the neural network) are fixed hyperparameters that do not scale with the input size. We added this information to Section 5.2 of the paper.
>
> The difficult step, which we prove to be NP-hard (see Appendix D), is the subset selection step. However, we use a fast approximation to make this problem feasible (see the last part of Algorithm 3).
>
> Didec improves on previous work by moving from an exponential to a polynomial algorithm, making the approach scalable.
>
> **Comment:** Severely Limited Scope and Insufficient Validation.
>
> **Answer:** We agree that having more and diverse domains can only benefit the work. So, we followed the reviewer's advice and included results for two Meta-World problems. Meta-World is a robotics environment with continuous actions and state spaces. We explain in Section 5.3.2 how to mask continuous input values and, in Section 7, how to simplify Didec's pipeline to accommodate continuous actions, given that the Levin loss optimization is inherently discrete. The core idea is to use the entire trajectory of the training problems to learn masks. That way, the number of options is small, and we bypass the discrete selection step.
>
> The experiments in Meta-World serve two purposes: to show that our mask-based decomposition can also be used in continuous spaces and in a simpler setting that bypasses the subset selection step.
>
> The results are shown at the bottom of Figure 2; similarly to the other domains, Didec demonstrates an advantage over the baselines evaluated.
>
> **Comment:** Stacked Heuristics.
>
> **Answer:** The problems we are dealing with (including the underlying RL problem) are computationally difficult (e.g., minimizing the Levin loss in our setting is NP-hard), so we have no alternative but to accept the use of heuristics as a way of scaling to larger problems.
>
> In our pipeline, we lose optimality as soon as we start using gradient descent to decompose neural networks. This is necessary to scale to larger problems, and we are aligned in spirit with Deep RL, which also relies on approximations.
>
> **Comment:** Authors need to work on the concise and accurate math derivations.
>
> **Answer:** We have all the key equations and pseudocode in the main text, which is all the pipeline needed for masking. The secondary structure (e.g., subset selection, hill-climbing search) is explained with equations and pseudocode in the appendix.
>
> We would be happy to accommodate any specific suggestions you may have for equations and pseudocode that we should move from the appendix to the main text.
>
> **Comment:** I acknowledge the improvements over Dec-Option of this work but raised my concerns as in weakness section. I do think the direction Dec-Option proposed is an exciting and novel direction worth to explore. But the claim authors made at the beginning "scaling Dec-Option up" turns to be a disappointment. I'm willing to raise my score if the author can prove their scalability to "real" problem such as locomotion or VLA tasks.
>
> **Answer:** We hope we have addressed your concerns with the newly added results. Please let us know if you have any other questions.

---

### Official Review · Reviewer_PkrE · 2025-11-05

**Soundness:** 3
**Presentation:** 2
**Contribution:** 3
**Rating:** 4
**Confidence:** 4

**Summary:**

This work introduces a hierarchical approach (called DIDEC) that reuses the set of learned policies and trajectories from previous tasks to define options for new tasks. This is mainly done by first converting the policies into fixed horizon options, then using them to discover reusable neural sub-functions by treating network decomposition as a *differentiable masking problem*. Building on the prior Dec-Options framework, which exhaustively enumerated neural sub-trees, DIDEC uses gradient-based learning to mask neurons and inputs, thereby identifying sub-functions and their “default parameters” that generalize across tasks. This enables option discovery in large networks and supports transfer and lifelong learning. Experiments across MiniGrid, MiniHack, and ComboGrid environments using A2C and PPO demonstrate that DIDEC improves sample efficiency and task generalization compared to various baselines.

**Strengths:**

* **Significance and originality:** The work contributes to both hierarchical and lifelong R by proposing a novel way to adapt learned policies for new tasks through learned masks.
* **Methodological depth:** Provides a detailed formulation, algorithms, and training procedure (with pseudocode and appendices). The approach generalizes across RL algorithms (A2C, PPO) and architectures (MLP, LSTM).
* **Experimental validation:** Demonstrates strong sample efficiency and transfer benefits across diverse domains. Particularly impressive is DIDEC’s ability to solve the 5×5 sparse-reward ComboGrid, where all baselines fail to converge.
* **Ablation studies:** Includes informative ablations showing the effects of input-only vs. neuron-only masking, illustrating the versatility of the method.
* **Clarity and completeness:** The paper is well structured and situates the work clearly within related fields of option discovery, masking, and programmatic RL.

**Weaknesses:**

- **Clarity:** The paper dedicates excessive space to background, delaying the introduction of the main method until Section 5 (page 4). This reduces focus on the novel contributions and experimental insights. This is potentially why the main algorithm (DIDEC) is only fully described in the appendix, which hinders readability and accessibility in the main text. Also the lifelong setup in only first mentioned in the background and the critical dependence on prior trajectories comes even much later. It would have helped if this context appeared in the abstract and introduction.

- **Lack of analysis of learned options:** The paper does not explore what the masked options learn. E.g., their termination sets, which inputs are masked, or how they perform when reused in original tasks after masking.

- **Missing failure analysis:** No discussion of when DIDEC fails, such as tasks with significantly different horizon lengths from prior ones (which may break deterministic termination assumptions), or when offline-learned masks face distribution shift in online fine-tuning (a common problem with offline RL). There is also no analysis of performance trends as the number of prior tasks increases.

- **Potential bias in seed selection:** Using fixed training/testing environment/task configurations (defined by handpicked seeds) without random resampling raises concerns about potential cherry-picking of favorable configurations. Sampling environment/task configuration seeds per run would yield a more robust evaluation.

- **Fairness and runtime concerns:** DIDEC requires an additional offline training phase using trajectories from prior tasks, raising questions of fairness in comparison to baselines and missing runtime comparisons. The experiments are also missing comparisons with other masking-based methods from related work (e.g., weight or input mask learning approaches), which would strengthen claims of novelty.

- **Minor:**
  - Line 87 notation ($π_j j = 1^{i−1}$) is unclear.
  - Figures (e.g., Figure 9) omit full visualizations for all baselines.
  - Definition of $d_{obs}$ per environment is missing (is it a flattened grid observation?)?

**Questions:**

1. Could the authors analyze the learned options (termination sets, input masks, average individual performance) to provide intuition about what behaviors DIDEC extracts? Similarly also analyse option usage during testing and final learned policies.
2. How does performance scale as the number of prior tasks increases?  Does DIDEC saturate or degrade with larger option libraries?
3. What are the failure modes? For example when previous tasks differ significantly in horizon or reward structure?
4. Can the authors assess runtime and computational overhead compared to baselines, given the additional offline mask training phase?
5. How sensitive are results to choice of training/testing seeds? Would randomizing them per run affect DIDEC’s performance consistency?
6. Could the paper include a comparison with prior masking-based transfer approaches to better position DIDEC’s advantages?
7. What steps could mitigate potential distribution shift from offline-learned masks to online adaptation?
8. Please clarify the ambiguous notation (e.g., line 87) and include complete visualization heatmaps for all baselines.
9. Is DIDEC robust to architectures or activations beyond ReLU/tanh, and how would it adapt to continuous input domains?

---

> ### Author Response · Authors · 2025-11-28
>
> **Question:** Could the authors analyze the learned options (termination sets, input masks, average individual performance) to provide intuition about what behaviors DIDEC extracts? Similarly also analyse option usage during testing and final learned policies.
>
> **Answer:** We have added an example of a trajectory from a test agent on ComboGrid (Appendix E), which shows that the test agent only uses learned options. Moreover, it shows that the options learned in this domain are quite modular and reusable.
>
> We have also added the exploration heatmaps for MiniGrid in Figure 13 in the Appendix. This plot shows the effect of the options Didec learns in terms of exploration.
>
> **Question:** How does performance scale as the number of prior tasks increases? Does DIDEC saturate or degrade with larger option libraries?
>
> **Answer:** We also added experiments on MultiRoom where we doubled the number of training problems used in our original experiments (see Figure 5 in Appendix F). The performance degrades slightly. Since the smaller training set is a subset of the larger one, the decrease in performance could be explained by the hill-climbing search being unable to find helpful subsets of options in the larger $\Omega$ set. This issue can potentially be mitigated by increasing the search budget in the subset selection step.
>
> **Question:** What are the failure modes? For example when previous tasks differ significantly in horizon or reward structure?
>
> **Answer:** The failure mode is when the training problems do not include the behaviors needed to learn helpful options. For example, if only one of the ComboGrid training problems included the combo to turn right, Didec would not be able to learn an option for turning right. Didec needs to find a sub-function in a policy that covers the behavior of another policy, so it needs at least two policies that encode the behavior. We added this discussion to the first paragraph of Appendix F.
>
> **Question:** Can the authors assess runtime and computational overhead compared to baselines, given the additional offline mask training phase?
>
> **Answer:** We added the running time for both MiniGrid and Meta-World (new domain). In MiniGrid, the running time of Didec is similar to Fine-Tune (~47 minutes for Didec and ~37 minutes for Fine-Tune), but with a much stronger performance. In Meta-World, we consider a simplified version of Didec, where we generate a small number of options so we bypass the selection step. In this simplified version, learning masks only takes a few seconds.
>
> Learning the masks and selecting options is much faster than learning the base policies. We always viewed the option learning process as part of the sleep cycle in wake-sleep algorithms. In the wake phase, the agent learns how to solve problems; in the sleep phase, the agent organizes what it has learned into helpful options. Please see Appendix K for the details.
>
>
> **Question:** How sensitive are results to choice of training/testing seeds? Would randomizing them per run affect DIDEC’s performance consistency?
>
> **Answer:** Didec could be sensitive to the choice of the training problems, as we explained in our ComboGrid example above (and also in Appendix F). However, besides ComboGrid, the choice of training environments was arbitrary. We further explore this with an additional experiment on MultiRoom, where the training environments are selected randomly. Didec is as sample-efficient in this setting as it was with the fixed training environments (see Figure 4 in Appendix F).
>
> **Question:** Could the paper include a comparison with prior masking-based transfer approaches to better position DIDEC’s advantages?
>
> **Answer:** Alikhasi & Leli (2024) showed that Modulating Masks (Ben-Iwhiwhu et al., 2022) perform poorly in the option-learning setting we consider in our own work. That is why we have not included it. We included only baselines that either performed well in previous experiments or align with Didec's learning process (e.g., the Fine-Tuned baseline). Did you have a specific baseline in mind?
>
> **Question:** What steps could mitigate potential distribution shift from offline-learned masks to online adaptation?
>
> **Answer:** This is an excellent question, and as far as we know, an open question. Didec's masking scheme can be quite effective in terms of handling distribution shifts, but more research is needed to understand how it can be used in an online setting.
>
> **Question:** Please clarify the ambiguous notation (e.g., line 87) and include complete visualization heatmaps for all baselines.
>
> **Answer:** In line 87 it should be the set of policies, which we fixed in the revised version. We have also added the heatmaps for all baselines. Finally, $d_{\text{obs}}$ is the flattened version of the input grids. This is clarified in Appendix D.

---

> > ### Author Response · Authors · 2025-11-28
> >
> > **Question:** Is DIDEC robust to architectures or activations beyond ReLU/tanh, and how would it adapt to continuous input domains?
> >
> > **Answer:** Didec's neuron masking requires the activation to be piecewise linear, which includes all functions from the ReLU family, such as Leaky ReLUs. As for input masking, we only tested it with ReLU and tanh, but we believe it would not work with any activation function.
> >
> > Regarding continuous input domains, we have conducted experiments on Meta-World, a robotics domain with continuous action and state spaces, demonstrating that Didec is also applicable to such domains. Input masking of continuous features is explained in Section 5.3.2. The results of Didec are shown in Figure 2. Didec is the only system evaluated to learn effective policies in this domain.

---

### Author Response · Authors · 2025-11-28
**Thank you!**

We thank all reviewers for their detailed and thoughtful feedback. We have incorporated their suggestions in the revised version, which we have uploaded onto OpenReview. All the modified text is marked in blue in the revised paper. As a summary of the changes we carried out, we have the following:

1. Described how Didec can be used to mask continuous inputs (Section 5.3.2) [PkrE and Nv3Y]
2. Added experiments on Meta-World, a robotics-like domain (Figure 2) [Nv3Y]
3. Added a section detailing the time complexity of Didec (Section 5.2) [Nv3Y]
4. Included examples of options learned (Appendix E) [PkrE and TdxZ]
5. Included empirical results when the training problems are selected randomly (Figure 4) [PkrE]
6. Included empirical results for larger sets of training problems  (Figure 5) [PkrE]
7. Added running time results (Appendix K) [PkrE]
8. Added heatmaps for all baselines (Figure 13) [PkrE]
9. Added a proof that selecting a subset of options that minimizes the Levin loss is NP-hard (Appendix D) [TdxZ]

The paper has improved significantly, thanks to your feedback. We look forward to answering any other questions you may have.

Best regards,
Authors

---

### Meta-Review · Area_Chair_mX7p · 2026-01-05

**Summary:**

The reviewers raised the following concerns:
- Limited novelty by being a follow up work.
- Limited contribution: The proposed algorithm is not a goal-driven option discovery framework but a supervised, brute-force behavior cloning system.
- Lack deep analysis: The paper does not explore what the masked options learn,  and how they perform when reused in original tasks after masking. There is also no discussion of when DIDEC fails.
- Lack theoretical justification and accurate math derivations.
- Fairness concern: Using fixed environments defined by handpicked seeds raises concerns about potential cherry-picking of favorable configurations. DIDEC requires an additional offline training phase using trajectories from prior tasks, raising questions of fairness in comparison to baselines.
- Computational Complexity: the paper does not provide runtime comparison with baselines. Current complexity of DIDEC is doubtful to be applied to actual RL envs.
- Limited Scope: it is unclear how the proposed method adapts to continuous input domains, how robust to activations beyond ReLU/tanh and distribution shifts.
- Insufficient Validation: Validation on grid-worlds alone is not sufficient. Standard benchmarks (MuJoCo, DMC, robotics) was not tested.
- Unclear scalability as the number of prior tasks increases.
- Stacked Heuristics: The method is not a principled solution. It relies on a proxy loss (cross-entropy) to generate options and a separate, non-optimal heuristic search (SHC) to select them. This stack of approximations lacks any formal guarantees.
- Not well organized: the main algorithm (DIDEC) is only fully described in the appendix, which hinders readability and accessibility in the main text.

**Reviewer Concerns:**

The following reviewer concerns have been addresses partially by the rebuttal:
- Computational Complexity: Added a section detailing the time complexity of Didec.
- Limited scope: Added experiments on Meta-World, a robotics-like domain.
- Lack deep analysis: Described how Didec can be used to mask continuous inputs.
- Fairness concerns

The concerns that are outstanding:
- Stacked Heuristics
- Limited novelty
- Lack deep analysis
- Scalability issue
- Lack theoritical justification and accurate math derivations.
- Not well organized

**Reviewer Scores:**

Reviewer PkrE might raise the score from 4 to 6 since some of his concerns were addressed by the rebuttal.

Reviewer Nv3Y might keep his negative score since the rebuttal did not fully addressed his concerns.

Reviewer TdxZ might keep his positive score after the authors addressed his two concerns.

[Note] This paper exceeds 9 page limit and should be desk-rejected.

---

### Decision · Program_Chairs · 2026-01-26

Reject